# On the Epistemic Limits of Personalized Prediction

**Lucas Monteiro Paes\***
Harvard SEAS
lucaspaes@g.harvard.edu

**Carol Xuan Long\***
Harvard SEAS
carol_long@g.harvard.edu

**Berk Ustun**
University of California, San Diego
berk@ucsd.edu

**Flavio P. Calmon**
Harvard SEAS
flavio@seas.harvard.edu

## Abstract

Machine learning models are often personalized by using *group attributes* that encode personal characteristics (e.g., sex, age group, HIV status). In such settings, individuals expect to receive more accurate predictions in return for disclosing group attributes to the personalized model. We study when we can tell that a personalized model upholds this principle for *every* group who provides personal data. We introduce a metric called the *benefit of personalization* (BoP) to measure the smallest gain in accuracy that any group expects to receive from a personalized model. We describe how the BoP can be used to carry out basic routines to audit a personalized model, including: (i) hypothesis tests to check that a personalized model improves performance for every group; (ii) estimation procedures to bound the minimum gain in personalization. We characterize the reliability of these routines in a finite-sample regime and present minimax bounds on both the probability of error for BoP hypothesis tests and the mean-squared error of BoP estimates. Our results show that we can only claim that personalization improves performance for each group who provides data when we explicitly limit the number of group attributes used by a personalized model. In particular, we show that it is impossible to reliably verify that a personalized classifier with $k \geq 19$ binary group attributes will benefit every group who provides personal data using a dataset of $n = 8 \times 10^9$ samples – one for each person in the world.

## 1 Introduction

Machine learning models often assign predictions on the basis of group attributes that denote personal characteristics. In medicine, for example, clinical prediction models account for characteristics that may be *protected* (e.g., sex), *sensitive* (e.g., HIV status), or *costly* to acquire (e.g., presence of a malignant tumor as determined via a biopsy) [3, 32]. Unlike lending or hiring, models that use personal characteristics in such applications are not subject to scrutiny due to an assumption that everyone would benefit from a more accurate model [7, 31, 33]. In an effort to develop models that are as accurate as possible, practitioners use every piece of information to maximize predictive accuracy over a heterogeneous population.

The prevalence of machine learning models that use personal characteristics reflects a belief that personalization will produce a uniform gain in performance across all groups. In effect, individuals who provide personal data to models expect more accurate predictions in return. In practice, however,

---

*Equal contribution.

36th Conference on Neural Information Processing Systems (NeurIPS 2022).

| Group | $n$ | $\hat{R}(h_p)$ | $\hat{R}(h_0)$ | $\hat{R}(h_0) - \hat{R}(h_p)$ | $n$ | $\hat{R}(h_p)$ | $\hat{R}(h_0)$ | $\hat{R}(h_0) - \hat{R}(h_p)$ |
|---|---|---|---|---|---|---|---|---|
| | | TRAINING DATA | | | | AUDITING DATA | | |
| `Female, W, NR` | 4,989 | 15.6% | 15.4% | -0.2% | 5,072 | 15.0% | 14.7% | -0.3% |
| `Female, W, NR` | 986 | 10.2% | 10.9% | 0.6% | 945 | 10.7% | 10.3% | -0.4% |
| `Male, W, NR` | 1,331 | 21.2% | 20.2% | -1.0% | 1,292 | 20.7% | 20.2% | -0.5% |
| `Female, W, R` | 409 | 16.1% | 15.6% | -0.5% | 399 | 16.0% | 15.3% | -0.8% |
| `Female, NW, R` | 265 | 21.1% | 22.3% | 1.1% | 248 | 20.6% | 19.4% | -1.2% |
| `Male, NW, R` | 587 | 25.0% | 23.0% | -2.0% | 667 | 24.4% | 21.6% | $-\textbf{2.8\%}^*$ |
| `Male, W, NR` | 15,012 | 21.7% | 22.0% | 0.2% | 14,939 | 21.1% | 21.5% | 0.4% |
| `Male, W, R` | 1,141 | 20.9% | 23.0% | 2.1% | 1,158 | 19.3% | 22.1% | 2.8% |
| **Total** | 24,720 | 20.0% | 20.1% | 0.1% | 24,720 | 19.3% | 19.5% | 0.2% |

**Table 1:** Personalized models may not assign more accurate predictions for every group who provides personal data. Here, we show this effect for a personalized classification task on the UCI Adult dataset [19] with three group attributes: Sex × Race × ImmigrationStatus ∈ {Female,Male} × {White,NonWhite} × {Resident,NonResident}. We fit a personalized logistic regression model with a one-hot encoding of the group attributes $h_p$. We then measure the gains to personalization with respect to a logistic regression model trained on a dataset without group attributes – i.e., the *generic model* $h_0$. Here, $\hat{R}(h_p)$ and $\hat{R}(h_0)$ are the error rates of $h_p$ and $h_0$ and $\hat{R}(h_0) - \hat{R}(h_p)$ is the gain from personalization. As shown, the personalized model $h_p$ improves overall accuracy but assigns less accurate predictions for specific groups (highlighted in orange). Moreover, these effects arise on the training dataset and the auditing dataset [see 31, for other examples]. Our proposed metric – the benefit of personalization – flags these instances by measuring the worst-case gain attain by any group (i.e., -2.8% for the group `Male, NW, R`).

models trained with personal data do not necessarily assign more accurate predictions to all those who provide it. Standard techniques for empirical risk minimization use personal characteristics to improve performance at a population level. However, these gains are not uniformly distributed across the groups who provide personal data. As shown in Table 1, personalization can improve performance for some groups while reducing performance for others. These instances of "worsenalization" violate the implicit promise of personalization: namely, that each group who provides personal data will receive a tailored performance improvement in return. In practice, they are prevalent, hard to detect, and hard to avoid – underscoring the need to measure the gains of personalization [31].

Practitioners must measure the gains of personalization whenever they train a personalized model. As shown in Table 1, one only needs to estimate performance gains relative to a *generic model* trained without group attributes, then report these gains for each group who provides personal data. Even as this audit is easy to carry out, it may not always produce clear recommendations. The estimated gains of personalization for a specific group can change drastically if we have too few samples for that group. In fact, it is not possible to detect if personalization helps or hurts at the group level – and by how much – without accounting for uncertainty. In practice, estimating the gains in personalization requires sufficient samples per group. When lacking, the reported gain may be just an illusion, a justification to collect personal information and use the personalized classifier that stems from statistical fluctuations due to limited samples per group or, equivalently, the use of too many features that encode personal characteristics.

In this paper, we study when we can verify if personalization works as it should – i.e., when a model that uses group attributes improves expected performance for every group who provides personal data. We derive information-theoretic bounds to characterize the reliability of testing and estimation routines in this setting. We then use these bounds to identify epistemic limits in personalization – i.e., conditions under which all testing and estimation routines are too unreliable to tell if personalization improves or worsens performance for each group who provides personal data. Our results can serve as rules of thumb to inform a range of stakeholders – be it model developers who wish to build models that reliably benefits all groups, to auditors who wish to verify that this is indeed the case.

The main contributions of this work include:

1. We introduce a metric – called the *benefit of personalization* (BoP) – to evaluate the quality of personalization in a model with group attributes. The BoP represents the minimum gain in accuracy that any group can expect from personalization. Measuring the BoP can flag instances where a personalized model aligns with our basic expectations of personalization. A positive BoP indicates that each group can expect to receive more accurate predictions in exchange for their personal information. In contrast, a negative BoP indicates that some groups who provide personal information can expect to receive worse predictions in return.

2. We design a hypothesis test to audit if a personalized model violates fair use. We pair our test with an information-theoretic limit on the error probability that holds for *any* hypothesis test that audits a model for fair use. A rule of thumb is that the number of samples per group should exceed the number of groups for testing error to be small.

3. We characterize the statistical limits on the Mean Squared Error (MSE) of any estimate of BoP. Our results show that we cannot reliably estimate the benefit of personalization across groups for models that use more than 25 group attributes without further assumptions on the data distribution.

**Related Work** We study the performance of machine learning models that assign predictions on the basis of personalized characteristics. Our goal is to flag personalized models that assign unnecessarily inaccurate predictions to groups who provide personal data. This differs from the underlying goal of existing measures of individual fairness [i.e., "treat similar individuals similarly"; 8] and group fairness [i.e., equalize performance across groups; 9, 13], which can be broadly viewed as measures to achieve *parity* [i.e., in that they seek to equalize predictions across individuals or equalize performance across groups 34]. Although parity is an appropriate notion of fairness in applications like lending and hiring, it is less suitable for applications with different ethical principles. For example, in medicine, the relevant principles are beneficence and non-maleficence [2]. In such settings, imposing parity may reduce performance for groups for whom the model performs well, rather than improving performance for groups for whom the model performs poorly [15, 22, 24, 25, 28]. In contrast, measuring and reporting the benefit of personalization can ensure that we use personal data in ways that improve performance for all groups without inflicting harm on specific groups.

Our results are related to a growing stream of work on auditing machine learning models [see e.g., 6, 18, 26, 27, 29, 30]. We consider a setting where we estimate the performance of a personalized model for a large number of intersectional groups known a priori [see e.g., 5, 10]. We characterize the reliability of this task when increasing the degree of personalization in a finite sample regime. Reliable verification becomes increasingly challenging in this setting. Given that number of groups grows exponentially with each additional group attribute, one needs exponentially more samples to reliably estimate the gains of personalization for each group. This is, of course, expected – and the motivation behind Hebert-Johnson et al. [14], Kearns et al. [17] who characterize the computational limits when auditing models with respect to parity-based metrics like equal opportunity and statistical parity under worst-case distributional assumptions. Here, we study the phenomenon in a setting where the number of samples is finite – e.g., because we are working with a dataset that is given [see e.g., 1, 16, 19]. In this way, we can delineate fundamental limits on how many group attributes can be used for personalization *before* testing and estimation become too unreliable (see e.g., Corollary 1 and Figure 1). What is *surprising* about our results is that, given finite sample, the *information-theoretic* maximum of group attributes that can be collected and still used to verify for a gain in personalization is, in fact, small: with more than 19 binary attributes, it is statistically impossible to test a personalized model for a gain in accuracy without further assumptions on the data distribution.

## 2 Measuring the Benefit of Personalization

In this section, we introduce formal definitions of fair use and the *benefit of personalization*.

### 2.1 Preliminaries

We start with an *auditing dataset* $\mathcal{D} \triangleq \{(\mathbf{x}_i, \mathbf{g}_i, y_i)\}_{i=1}^n$ of i.i.d. samples from a data distribution $\mathrm{P}_{\mathbf{X}, \mathbf{G}, Y}$. Each sample consists of a vector of features $\mathbf{x}_i \in \mathcal{X}$ (e.g. height, weight), a vector of *group attributes* $\mathbf{g}_i \in \mathcal{G}$ (e.g. sex, HIV status), and a class label $y_i \in \mathcal{Y}$ (e.g. incidence of stroke). Group attributes partition the dataset into *groups* of individuals with the same personal characteristics $\mathbf{g}_i = \mathbf{g}$. We use $\mathbf{g}$ to denote a group, and use $\mathrm{P}_{\mathbf{g}} \triangleq \mathrm{P}_{\mathbf{X}, Y \mid \mathbf{G} = \mathbf{g}} \ \forall \mathbf{g} \in \mathcal{G}$ to denote its conditional data distribution.

Assume that we are given a *personalized classifier* that uses group attributes $h_p : \mathcal{X} \times \mathcal{G} \to \mathcal{Y}$ and a *generic classifier* $h_0 : \mathcal{X} \to \mathcal{Y}$ that does not. We assume that these models are trained on a training dataset that is independent of the auditing dataset $\mathcal{D}$. The group risk (or loss) of the classifiers for group $\mathbf{g}$ with respect to the loss function $\ell : \mathcal{Y} \times \mathcal{Y} \to \mathbb{R}$ is defined as:

$$R(h, \mathbf{g}) \triangleq \begin{cases} \mathbb{E}\left[\ell(h(\mathbf{X}), Y) \mid \mathbf{G} = \mathbf{g}\right] & \text{if} \quad h : \mathcal{X} \to \mathcal{Y}, \\ \mathbb{E}\left[\ell(h(\mathbf{X}, \mathbf{g}), Y) \mid \mathbf{G} = \mathbf{g}\right] & \text{if} \quad h : \mathcal{X} \times \mathcal{G} \to \mathcal{Y}, \end{cases} \tag{1}$$

The empirical risk $\hat{R}(h, \mathbf{g})$ for each group is defined as:

$$\hat{R}(h, \mathbf{g}) \triangleq \begin{cases} \frac{1}{n_\mathbf{g}} \sum_{i:\mathbf{g}_i=\mathbf{g}} \ell(h(\mathbf{x}_i), y_i) & \text{if} \quad h : \mathcal{X} \to \mathcal{Y}, \\ \frac{1}{n_\mathbf{g}} \sum_{i:\mathbf{g}_i=\mathbf{g}} \ell(h(\mathbf{x}_i, \mathbf{g}), y_i) & \text{if} \quad h : \mathcal{X} \times \mathcal{G} \to \mathcal{Y}. \end{cases} \tag{2}$$

where $n_\mathbf{g}$ refers to the number of samples in group $\mathbf{g}$.

Individuals who provide their personal characteristics to a personalized classifier expect to receive a tailored gain in performance in return. In Definition 1, we frame these expectation in terms of collective preference guarantees that ensure *fair use* of group attributes [see 31, 33]

**Definition 1** (Fair Use of Group Attributes). The personalized classifier $h_p : \mathcal{X} \times \mathcal{G} \to \mathcal{Y}$ ensures the *fair use* of group attributes for group $\mathbf{g} \in \mathcal{G}$ with respect to the loss function $\ell : \mathcal{Y} \times \mathcal{Y} \to \mathbb{R}$ if

$$R(h_p, \mathbf{g}) \leq R(h_0, \mathbf{g}), \tag{3}$$

$$\mathbb{E}\left[\ell\left(h_p(\mathbf{X}, \mathbf{g}), Y\right) \mid \mathbf{G} = \mathbf{g}\right] \leq \mathbb{E}\left[\ell\left(h_p(\mathbf{X}, \mathbf{g}'), Y\right) \mid \mathbf{G} = \mathbf{g}\right] \ \forall \mathbf{g}, \ \mathbf{g}' \in \mathcal{G}. \tag{4}$$

Here, conditions in (3) and (4) reflect the collective preference guarantees of *rationality* and *envy-freeness*, respectively. These conditions capture the minimal expectations of a group from a personalized model in applications where groups prefer more accurate models. Given a personalized classifier that satisfies the rationality condition (3), the average person in group $\mathbf{g}$ would prefer the predictions from a personalized classifier $h_p$ to those generic classifier $h_0$. Moreover, given a personalized classifier that satisfies the envy-freeness condition (4), the average person in group $\mathbf{g}$ would prefer the predictions personalized for their group over predictions personalized for any other group $\mathbf{g}' \neq \mathbf{g}$. The paper focuses on ensuring the rationality condition as it reflects minimal conditions for personal characteristics to be used in a classification task.

The generic classifier $h_0 \in \mathcal{H}$ can be the empirical risk minimizer over a training dataset $\mathcal{D}_{\text{train}}$, i.e.,

$$h_0 \in \arg \min_{h \in \mathcal{H}} \sum_{(\mathbf{x}, \mathbf{g}, y) \in \mathcal{D}_{\text{train}}} \ell(h(\mathbf{x}), y),$$

where $\mathcal{H}$ is a family of classification models (e.g., linear classifiers, random forests). Equivalently, $h_p \in \mathcal{H}$ can represent the personalized model that minimizes empirical risk over a training dataset that includes group attributes. In this case, the auditing dataset $\mathcal{D}$ (independent of $\mathcal{D}_{\text{train}}$) can be used to empirically test whether $(h_0, h_p)$ guarantees fair use.

**Example.** Consider a classification task where the goal is to predict if a patient suffering from cancer will experience a remission. In this setting, a generic classifier $h_0$ may only use features that are readily available in a electronic health record. A personalized classifier $h_p$ may use these same features along with the group attributes $\mathbf{G} = \texttt{TumorType} \in \{\texttt{A}, \texttt{B}, \texttt{C}\}$. Here, a personalized model that violates the rationality condition for a group, e.g. $\mathbf{g} = (\texttt{B})$, would mean that patients may obtain more accurate predictions without undergoing an invasive procedure. In this case, patients should be informed of this fact when weighing the decision to provide more data. Moreover, a doctor may choose to only perform such invasive procedure when it is expected to produce a more accurate diagnosis *regardless of the result of the biopsy*.

## 2.2   The Benefit of Personalization

The rationality condition that fair use enshrines is not necessarily met when models are trained using empirical risk minimization. When rationality is violated, personalization may improve performance *across all groups*, yet reduce performance *for specific groups* [see Table 1 and 31]). Ideally, personalization lead to a gain – or at least not a loss – in performance regardless of the value of the group attribute $\mathbf{g}$. When this is not the case, stakeholders must be informed that providing more personal information may not necessarily lead to (average) accuracy gains.

To quantify the gain in performance across all groups, we introduce a mathematical definition for the *benefit of personalization* (BoP). This metric is a function of the personalized classifier $h_p$, the generic classifier $h_0$, and the data distribution. The BoP captures the minimum change in loss from using a personalized classifier across all intersectional groups defined by $\mathbf{g} \in \mathcal{G}$. Equivalently, the BoP quantifies the potential violation of the Rationality constraint in Definition 1.

**Definition 2** (Benefit of Personalization). Given a fixed data distribution $P_{\mathbf{X},\mathbf{G},Y}$ and generic model $h_0$, the *benefit of personalization* of a personalized model $h_p$ relative to generic model $h_0$ is:

$$\gamma(h_0, h_p) \triangleq \min_{\mathbf{g} \in \mathcal{G}} R(h_0, \mathbf{g}) - R(h_p, \mathbf{g}) \tag{5}$$

We refer to $R(h_0, \mathbf{g}) - R(h_p, \mathbf{g})$ as the the per-group benefit for group $\mathbf{g}$, and write $\gamma = \gamma(h_0, h_p)$ when the parameters are clear from context.

When the benefit of personalization is positive but small, individuals may prefer to receive predictions from a generic model $h_0$ rather than a personalized model $h_p$. This would reflect settings where the expected gain in personalization would be insufficient to warrant the cost of collecting or disclosing personal information (e.g., when the information is costly or sensitive). Testing for the benefit of personalization (e.g., if $\gamma > \epsilon$) is, on its own, insufficient: stakeholders must be made aware of the *estimated* gain in accuracy from reporting personal data and only provide their personal data when it leads to a sufficient gain in accuracy [31, 33]. However, calculating the BoP *exactly* requires knowledge of the data distribution (see Def. 2) – a condition rarely met in practice. Therefore, it is critical to understand how to estimate the BoP given an auditing dataset $\mathcal{D}$, as well as when this estimation may be infeasible given a limited sample size or a large number of group attributes.

## 2.3 Assumptions

In the remainder of this work, we restrict our attention to classification tasks with a binary outcome $\mathcal{Y} = \{0, 1\}$, and $k$ binary group attributes $\mathbf{G}_i \in \mathcal{G} \triangleq \{0, 1\}^k$ that specify $d \triangleq |\mathcal{G}| = 2^k$ groups. We assume that we work with an auditing dataset $\mathcal{D}$ that contains $m \triangleq \lfloor n/d \rfloor$ samples per group. This reflects an optimistic assumption in settings without prior knowledge of the data distribution $P_{\mathbf{X},\mathbf{G},Y}$. In effect, measuring the benefit of personalization is more unreliable in tasks with a disparate number of samples per group (see Appendix C.3).

We consider a setting where the benefit of personalization measures the minimum gain in *accuracy* between a personalized classifier and its generic counterpart. This corresponds to selecting the 0-1 loss function $\ell(y, h(\mathbf{x}, \mathbf{g})) \triangleq \mathbf{1}[y \neq h(\mathbf{x}, \mathbf{g})]$ in (2). The resulting quantity can be expressed as follows:

$$\gamma(h_0, h_p) = \min_{\mathbf{g} \in \mathcal{G}} \Pr(Y = h_p(\mathbf{X}, \mathbf{g}) \mid \mathbf{G} = \mathbf{g}) - \Pr(Y = h_0(\mathbf{X}) \mid \mathbf{G} = \mathbf{g}),$$

where $\Pr(Y = h_p(\mathbf{X}, \mathbf{g}) \mid \mathbf{G} = \mathbf{g}) - \Pr(Y = h_0(\mathbf{X}) \mid \mathbf{G} = \mathbf{g})$ is the gain in accuracy for group $\mathbf{g}$. Our results can be generalized to settings where the BoP measures the minimum gain using other performance metrics, such as the false positive rate or false negative rate (see Supplementary Material Section B.1).

Naturally, one would expect the personalized classifier $h_p$ and its generic counterpart $h_0$ to be trained in "good faith," i.e., they are not designed to underperform on a specific group. The BoP would not capture, for example, the case where both the personalized classifier *and* the generic classifier underperform on a specific group – it only reflects the *relative* gain in accuracy for each group by switching to a personalized classifier. In this regard, the BoP complements existing group fairness metrics.

## 3 Testing with the Benefit of Personalization

In this section, we describe a hypothesis testing framework to check that a personalized model produces a sufficiently large gain in performance for all groups. We then present an information-theoretic bound on the reliability of this procedure for all hypothesis tests.

### 3.1 Hypothesis Testing Setup

In Definition 3, we present a one-sided hypothesis test to verify if a personalized model improves expected accuracy for every group who provides personal data.

**Definition 3** (Hypothesis Test for Fair Use). Given a personalized classifier $h_p$, a generic classifier $h_0$, and auditing dataset $\mathcal{D}$, we verify whether $(h_p, h_0)$ yields an $\epsilon > 0$ gain in expected accuracy for

every group using a hypothesis test where:

$$H_0: \quad \gamma < 0 \quad \Leftrightarrow \quad \min_{\mathbf{g} \in \mathcal{G}} \quad R(h_0, \mathbf{g}) - R(h_p, \mathbf{g}) < 0$$

$$H_1: \quad \gamma \geq \epsilon \quad \Leftrightarrow \quad \min_{\mathbf{g} \in \mathcal{G}} \quad R(h_0, \mathbf{g}) - R(h_p, \mathbf{g}) \geq \epsilon$$

Here, the null hypothesis $H_0$ holds when a personalized model $h_p$ assigns predictions that are less accurate than those of a generic model $h_0$ for one or more groups. In contrast, the alternative hypothesis $H_1$ holds when a personalized model $h_p$ assigns predictions that produce an expected gain in accuracy of $\epsilon$ for each group.

In principle, the value of $\epsilon$ should be set as the minimal gain in expected accuracy that every group should receive from a personalized model. Setting $\epsilon = 0$ would flag a personalized model that assigns unnecessarily inaccurate predictions to at least one group. Setting $\epsilon$ to a larger positive value would flag a personalized model where at least one group experiences a gain that is too low to warrant the cost of providing personal information.

The hypotheses can be tested using a range of procedures, including a bootstrap test [12, 23] and a McNemar test [20]. Arguably the simplest procedure is a *threshold test*, which would require estimating the BoP as shown in Table 1 and reject $H_0$ if $\hat{\gamma} > \epsilon$.

In what follows, we derive a bound on the reliability of the hypothesis test in Def. 3 that holds for *all* procedures. For clarity of exposition, we represent the inputs and output for this hypothesis test as a *decision function* $\Psi : (\mathcal{D}, h_p, h_0, \epsilon) \rightarrow \{0, 1\}$, where $\Psi(\mathcal{D}, h_p, h_0, \epsilon) := \mathbf{1}[\text{Accept } H_1]$. Even as the setup above assumes that $\epsilon > 0$, our results will apply to settings where $\epsilon \leq 0$. When $\epsilon < 0$, our results will characterize the reliability of an inverted setup with $H_0 : \gamma > 0$ and $H_1 : \gamma \leq \epsilon$. The parameter $\epsilon$ is set to a value that is arbitrarily close to 0 – but not 0 – since the reliability of the hypothesis test requires separation between the regions $H_0$ and $H_1$.

## 3.2   Impossibility Results for Reliable Testing

We characterize the reliability of hypothesis tests in terms of their *probability of error*, defined below.

**Definition 4** (Probability of Error). Consider auditing a personalized classifier by testing the hypotheses $H_0$ and $H_1$ in Def. 3. Given a personalized classifier $h_p$, generic classifier $h_0$, auditing dataset $\mathcal{D}$, and threshold gain $\epsilon > 0$, the *probability of error* of a hypothesis test $P_e$ is defined as:

$$\mathrm{P}_e := \frac{1}{2} [ \underbrace{\Pr\left(\Psi(\mathcal{D}, h_p, h_0, \epsilon) = 1 \mid H_0 \text{ is True}\right)}_{\text{Type-I Error}} + \underbrace{\Pr\left(\Psi(\mathcal{D}, h_p, h_0, \epsilon) = 0 \mid H_1 \text{ is True}\right)}_{\text{Type-II Error}} ]$$

The probability of error captures both the false positive rate and the false negative rate of a given hypothesis test. If the probability of error of a hypothesis test exceeds 50%, then this means that a hypothesis test is as reliable as the outcome of an unbiased coin toss – too unreliable to allow for verification. The next theorem – and the main result in this section – states a *lower bound* on the probability of error when testing if the BoP exceeds $\epsilon$.

**Theorem 1** (Lower Bound on Probability of Error). *Consider auditing a personalized classifier $h_p$ in comparison to a generic classifier $h_0$ on a dataset $\mathcal{D}$ drawn from an (unknown) distribution with $d$ groups, $m = \lfloor n/d \rfloor$ samples per group. Moreover, let $P_e$ denote the probability of error for a hypothesis test as in Def. 4 with fixed $\epsilon \in [-0.5, 0.5]$. The worst-case probability of error over all pairs of data distributions $(P_{\mathbf{X}, \mathbf{G}, Y}, Q_{\mathbf{X}, \mathbf{G}, Y})$ where one distribution satisfies $H_0$ and the other $H_1$ – denoted by $P_{\mathbf{X}, \mathbf{G}, Y} \in H_0$ and $Q_{\mathbf{X}, \mathbf{G}, Y} \in H_1$, respectively – and all decision functions $\Psi : (\mathcal{D}, h_p, h_0, \epsilon) \rightarrow \{0, 1\}$ satisfies*

$$\min_{\Psi} \max_{\substack{P_{\mathbf{X}, \mathbf{G}, Y} \in H_0 \\ Q_{\mathbf{X}, \mathbf{G}, Y} \in H_1}} P_e \geq 1 - \frac{1}{\sqrt{d}} \left(1 + 4\epsilon^2\right)^{m/2}. \tag{6}$$

The minimax lower bound on the probability of error in Theorem 1 represents the worst-case performance of *any* hypothesis test without further assumptions on the data distribution. The minimax lower bound implies that for every decision function $\Psi$, there exists a pair of distributions – one satisfying $H_0$ and the other $H_1$ – such that the probability of error for the hypothesis test will exceed the right-hand side of (6).

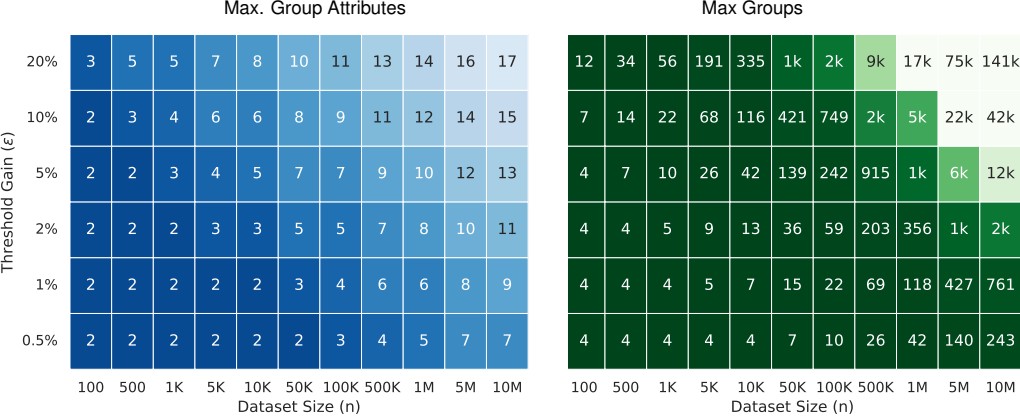

**Figure 1:** Epistemic limits of personalization for binary classification tasks with group attributes. We identify limits on the number of group attributes (left) and number of groups (right) to reliably test if a personalized classifier provides an expected gain of accuracy greater than $\epsilon$ gain for every group using a dataset with $n$ points. The limits correspond to largest values needed to ensure that $\min \max P_e \leq 1/2$ under Theorem 1, and are computed using the same technique as Corollary 1 (see Appendix B.2).

**Proof Sketch.** The proof of Theorem 1 constructs the lower bound for the probability of error for **any** decision function $\Psi$ by bounding the total variation distance of two distributions (belonging to different hypotheses) that are "difficult" to distinguish. We provide the full proof in the supplementary material Section B.1.

Theorem 1 characterizes the relationship between probability of error and the number of group attributes for personalization in settings with a finite number of samples. Intuitively, as the number of groups becomes larger, we have fewer samples per group, and the likelihood of erring in the hypothesis test increases. Using the minimax lower bound in Theorem 1, we can delineate a fundamental limit on the maximum number of group attributes that we can include in a personalized model to test if every group benefits from personalization without making further assumptions on the data distribution. In Corollary 1, we highlight this limit for an idealized scenario where the dataset $\mathcal{D}$ contains a sample for each person on Earth. In Figure 1 we highlight the maximum number of binary group attributes (left) and groups (right) for multiple dataset sizes and threshold gains.

**Corollary 1** (Epistemic Limit in Number of Group Attributes). *Consider auditing a personalized classifier $h_p$ to test if it provides a gain of at least $\epsilon = 0.01$ to each group on an auditing dataset $\mathcal{D}$ with $n = 8 \times 10^9$ samples – i.e., one for each person on Earth. If the personalized classifier uses more than $k \geq 19$ binary group attributes, then for any hypothesis test there will exist a pair of probability distributions $P_{\mathbf{X},\mathbf{G},Y} \in H_0, Q_{\mathbf{X},\mathbf{G},Y} \in H_1$ for which the test attains a probability of error that exceeds 50%, i.e.,*

$$k \geq 19 \implies \min_{\Psi} \max_{\substack{P_{\mathbf{X},\mathbf{G},Y} \in H_0 \\ Q_{\mathbf{X},\mathbf{G},Y} \in H_1}} P_e \geq \frac{1}{2}.$$

**Phase Transition** Theorem 1 characterizes the relationship between the minimax probability of error in testing $P_e$ in terms of the threshold gain $\epsilon$, the number of groups $d$, and data points per group $m$. This relationship highlights a phase transition in the reliability of testing. In particular, we find that probability of error exceeds $1/2$ whenever the number of samples per group exceeds the number of groups (see Figure 2). In the auditing setup in Corollary 1, the minimax lower bound on $P_e$ jumps from 13% to 97% as we increase the number of binary group attributes from $k = 18$ to $k = 19$. These phenomena highlight an informative rule of thumb – i.e., that testing is fundamentally unreliable in settings where the number of groups exceeds the number of samples per group.

**Epistemic Limits.** In Corollary 1 we have shown that for a fixed dataset size $\left(n = 8 \times 10^9\right)$ and a threshold gain ($\epsilon = 0.01$) there is a a limit on the number of binary group attributes to achieve reliable testing. In Figure 1 (left), we generalize this result and show how the maximum number of

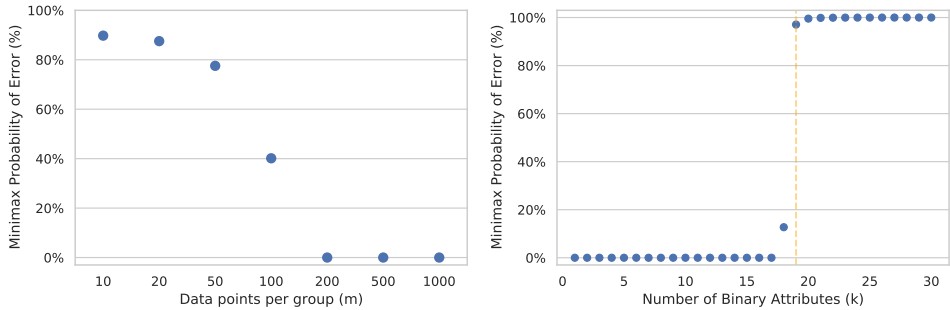

**Figure 2:** Minimax lower bound for the probability of error in hypothesis test. Illustration of the minimax lower bound of probability of error in Theorem 1 (left). For a fixed number of groups $d = 141$ and $\epsilon = 0.01$, when the number of samples per group decreases the probability of error in hypothesis test goes to $100\%$. We have, in the y-axis, $\max(0, \text{lower bound in } 6)$. Illustration of the minimax lower bound of probability of error in Theorem 1 (right). For a fixed sample size $n = 8 \times 10^9$ (one sample per person in the world) and varying the number of binary attributes $(k)$, there is a phase transition for the minimax error probability in a hypothesis test in $k = 19$ (orange line) as predicted in Corollary 1.

binary attributes for reliable testing changes with the dataset size and threshold gain. Remarkably, without any further assumption, for a dataset with at most 50k samples (usual in domains such as medicine) and $\epsilon = 0.01$ (just $1\%$ accuracy gain in personalization), at most 3 binary group attributes can be used in order to ensure reliable auditing for the BoP for any data distribution. The limits in Figure 1 are violated by *many* algorithms used in medicine (e.g, [4, 11] ). In Figure 1 (right) we drop the assumption of binary group attributes and show the maximum number of groups to ensure reliable testing. Note that, in general, the number of groups when we drop the binary attributes assumption is bigger than the number of groups generated by the binary group attributes. This is an effect from rounding – i.e., when $2^{k_{\max}+1} > d_{\max}$ but $2^{k_{\max}} < d_{\max}$, where $k_{\max}$ is the maximum number binary group attributes and $d_{\max}$ is the maximum number of groups to ensure reliable testing.

We reiterate that the error bounds in Theorem 1 and Corollary 1 hold in the minimax sense. These bounds are conservative estimates that may be relaxed given further assumptions on the data distribution $P_{\mathbf{X},\mathbf{G},Y}$. Moreover, the test in Def. 3 considers only two hypotheses for a pre-specified $\epsilon$. Hence, this testing procedure assumes *a priori* that the true BoP $\gamma$ cannot satisfy $0 < \gamma < \epsilon$. In practice, such prior information about the BoP may not be known, requiring $\gamma$ to be estimated directly from an auditing dataset $\mathcal{D}$.

## 4 Estimating the Benefit of Personalization

In this section, we characterize information-theoretic limits when estimating the benefit of personalization. We identify a threshold number of group attributes beyond which, in general, it is impossible to estimate the BoP reliably.

### 4.1 Estimation Setup

We consider the problem of designing an estimator $\gamma^*$ for the BoP for a dataset $\mathcal{D}$. Our metric of choice is the *mean-square error* (MSE) $\mathbb{E}\left[(\gamma - \gamma^*)^2\right]$. Our goal is to characterize the minimal MSE (MMSE) across all possible data distributions – i.e., without further assumptions on the data distribution. In particular, corresponding bounds allow practitioners to design a dataset collection that ensures a priori reliable estimation of the BoP. Therefore, we analyze the minimax estimation error – i.e., the best MMSE achievable across *all* data distributions, defined as

$$\Delta(h_0, h_p) \triangleq \inf_{\gamma^* \in \sigma(\mathcal{D})} \sup_{P_{\mathbf{X},\mathbf{G},Y}} \mathbb{E}\left[(\gamma - \gamma^*)^2\right], \tag{7}$$

where $\gamma^* \in \sigma(\mathcal{D})$ denotes that $\gamma^*$ is measurable with respect to the distribution $P_{\mathbf{X},\mathbf{G},Y}^n$ of $\mathcal{D}$.

To further motivate the definition of $\Delta$, suppose that there exist functions $g_1(d, m), g_2(d, m) > 0$ such that $\Delta \leq g_1(d, m)$, and $\Delta \geq g_2(d, m)$. Recall that $d$ is the number of groups and $m$ is the number of samples per group. This implies that for every estimator $\gamma^*$, there exists a data distribution

$P_{\mathbf{X},\mathbf{G},Y}$ such that $g_2(d,m) \leq \mathbb{E}\left[(\gamma - \gamma^*)^2\right] \leq g_1(d,m)$. In practice, this means that for *any* estimator there is a distribution such that the MSE of the resulting BoP estimate will be bounded away from zero by $g_2(d,m)$. The main result in this section is Theorem 2, which gives explicit expressions for $g_1, g_2$.

For a dataset with a fixed amount of samples ($n$), when the number of group attributes increases, the number of data points per group decreases. Intuitively, the estimation variance will be higher on the "more" personalized classifier because there are fewer samples per group. Theorem 2 gives a characterization for the order of the estimation error for the BoP.

**Theorem 2** (Minimax bounds for the Estimation MSE). *Given a dataset $\mathcal{D}$, constants $d, m \in \mathbb{N}$ such that $4 \leq d \leq 2^{m+2}$, a generic ($h_0$) and a personalized ($h_p$) classifier, the minimax estimation error for the benefit of personalization in* (7) *is such that:*

$$\frac{\log(d)}{16m} - \frac{\log(4)}{16m} \leq \Delta(h_0, h_p) \leq \frac{\log(d)}{m} + \frac{\log(m)}{m} + \frac{2 + \log(2)}{m}, \tag{8}$$

**Proof Sketch.** The proof of Theorem 2 relies on two-point methods for minimax lower bounds. This result uses a generalized version of Le Cam's Lemma stated in Lemma 1 in the supplementary material (which is of independent interest) together with standard concentration inequalities. We provide the detailed proof in the supplementary material and discuss its implications in Section 4.2.

The bounds in Theorem 2 are illustrated in Figure 3 (right) in $\log$ scale. This figure suggests that even though the bounds in Theorem 2 are in the minimax sense, they are close to the estimated MSE, especially the lower bound. In addition, in Figure 3 the minimax lower bound is *not* a lower bound for the MSE of the BoP estimation. This phenomenon is expected and is a consequence of the generality o the minimax bounds that hold for all decision functions and estimators and inherently consider the worst-case distribution for each case. Additional knowledge of the underlying data distribution may result in a more favorable trade-off between sample size, estimation error, and the number of group attributes.

## 4.2 Implications of the Limits of Estimating the BoP

Theorem 2 delineates a fundamental limit on the number of group attributes to estimate the BoP reliably. The lower (converse) bound in (8) captures the trade-off between the minimax MSE of an estimator, the number of group attributes, and the size of the dataset. In Corollary 2, we express this trade-off by identifying an upper bound for the number of group attributes required to achieve a precise (in the minimax MSE sense) BoP estimate for a fixed sample size.

**Corollary 2** (Limits of Reliable Estimation). *Given a dataset $\mathcal{D}$ with $n$ samples, $d \geq 4$ groups, $m = n/d$ samples per group, $0 < \delta < 1/16$, and a generic and a personalized classifier. If $d \leq 2^{m+2}$, and $\Delta(h_0, h_p) \leq \delta$, the number of binary group attributes ($k$) satisfies,*

$$(k-2)2^k \leq \frac{16}{\ln 2} n\delta. \tag{9}$$

**Remark.** *For $n = 8 \times 10^9$ (i.e., the approximate number of people on earth) and $\delta = 0.05^2$ (i.e., at most 5% MSE), then one can only reliably estimate the BoP for a personalized model with at most $k \leq 25$ binary group attributes.*

The proof of Corollary 2 is a direct application of the Theorem 2 (see Appendix B.5 for the proof). Setting $\delta = 0.05^2$ caps the square root of the MSE to 5%. In settings where the estimator $\gamma^*$ is unbiased, the result in Corollary 2 caps the variance of the estimator to at most 5%.

Corollary 2 gives a limit in the number of binary group attributes in order to ensure a $\delta$-*precision* for the error in BoP estimation – i.e., to guarantee that $\Delta(h_0, h_p) \leq \delta$. Eq. 9 can help practitioners decide the number of group attributes used by $h_p$ by (i) first setting a target estimation precision $\delta$ for the BoP, where $0 < \delta < 1/16$, then (ii) determining the maximum number of group attributes by computing $k_{\max} = \max\{k \in \mathbb{N}; (k-2)2^k \leq \frac{16}{\ln 2} n\delta\}$. Note that, since our bounds are minimax, this renders a conservative estimate on the number of attributes used for personalization.

In general, one may expect that the maximum number of group attributes for a reliable test should exceed the maximum number of group attributes for reliable estimation. Our results in (6) and (8) show otherwise. This is an artifact of the fact that $|\gamma| \in [0,1]$. Easier estimation for statistics with norm smaller than 1 is simply an artifact of scale and follows from the Le Cam Lemma [21].

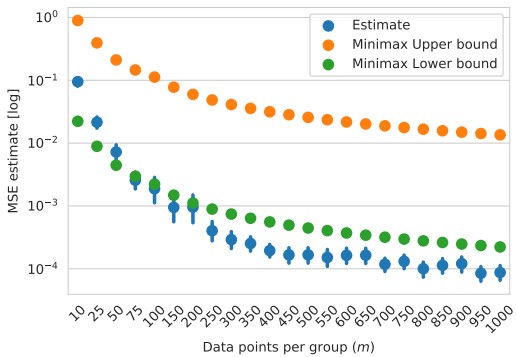

**Figure 3:** Overview of estimation error for the BoP of personalized classifiers on a semi-synthetic dataset built from the UCI Adult dataset [19]. We plot the MSE of BoP estimates as we increase the number of samples per group, along with the minimax upper and lower bounds from Theorem 2 in log-scale. Here, we compute each estimate using 100 Monte Carlo iterations, and show $95\%$ confidence intervals estimated via bootstrap. As shown, the MSE decreases exponentially as the number of samples per group increases. The minimax bounds are close to the estimated values, especially the lower bound, which indicates that the worst-case data distribution considered in the minimax bound is approximates the empirical data distribution. The generality of the bounds implies that some knowledge of the underlying data distribution may result in a more favorable trade-off between sample size, estimation error, and the number of group attributes. Here, the MSE in the BoP estimate falls below the minimax lower bounds, suggesting that the model's underlying distributions for each group are not quite the worst-case scenario.

## 5   Concluding Remarks

Personalized models should improve performance for all individuals who provide personal data. In this work, we introduced a measure called the *benefit of personalization* (BoP) to verify this basic principle for classification models that are personalized using group attributes. Using the BoP, we characterized the reliability of testing and estimation procedures to verify this principle in practice. We then determined conditions under which testing and estimation were too unreliable to guarantee gains across groups. Our results highlight fundamental limits in our ability to tell if personalization leads to gains in finite-sample settings. In particular, we show that there is no way to reliably determine if a personalized model with $k \geq 19$ attributes benefits all groups who provide personal data on a dataset with $n = 8 \times 10^9$ samples (i.e., one per person in the world) without making further assumptions on the data distribution. These limits can guide how to define and select group attributes to ensure reliable testing and estimation, allowing practitioners to make informed decisions when personalizing models with characteristics that are protected, sensitive, or costly to acquire.

**Acknowledgements.**   This material is based upon work supported by the National Science Foundation under grants CAREER 1845852, IIS 1926925, and FAI 2040880.

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
