# Supplementary Material:
# On the Epistemic Limits of Personalized Prediction

**Lucas Monteiro Paes\***
Harvard SEAS
lucaspaes@g.harvard.edu

**Carol Xuan Long\***
Harvard SEAS
carol_long@g.harvard.edu

**Berk Ustun**
University of California, San Diego
berk@ucsd.edu

**Flavio P. Calmon**
Harvard SEAS
flavio@seas.harvard.edu

## A Preliminaries and Notation

**Preliminaries.**  In the supplementary material we provide the following information.

- Appendix B provides mathematically rigorous proofs of the theorems, corollaries, and lemmas in the main paper,
- Appendix C further discusses the interpretation of minimax bounds, the assumptions made in the proof, and how the bounds in the main paper can be used.
- Appendix D gives more details for the experiments in the main paper and provides additional numerical on two datasets (HSLS (Ingels et al., 2011) and COMPAS (Angwin et al., 2016) ).

**Notation Table.**

| Symbol | Meaning |
|--------|---------|
| $\mathcal{D}$ | dataset |
| $P_{\mathcal{D}}$ | distribution of the dataset |
| $\mathcal{X}$ | non-group features space |
| $\mathcal{G}$ | group features space |
| $h_0$ | group blind classifier |
| $h_g$ | personalized classifier |
| $\gamma$ | BoP |
| $\gamma^*$ | estimator of BoP |
| $n$ | total number of samples |
| $d$ | number of groups |
| $m$ | samples per group |
| $k$ | number of group attributes |
| $P_e$ | probability of error of a hypothesis test |
| $\Psi$ | decision function |
| $\epsilon$ | threshold in hypothesis test |

**Table 1:** Notation

**Code**  For the code used in all experiments of this paper visit the link: `https://github.com/LucasMonteiroPaes/On-the-Epistemic-Limits-of-Personalized-Prediction.git`.

36th Conference on Neural Information Processing Systems (NeurIPS 2022).

# B Proofs

In this section we provide proofs for all the theorems and corollaries in the main paper. For clarity of exposition, we re-state the result before each proof. Note that the proofs use assumptions stated in the main text.

## B.1 Proof of Theorem 1

**Theorem 1** (Lower Bound on Probability of Error). *Consider auditing a personalized classifier $h_p$ using a generic classifier $h_0$ on a dataset $\mathcal{D}$ drawn from an (unknown) distribution $P_{\mathbf{X},\mathbf{G},Y}$ with $d$ groups and $m = \lfloor n/d \rfloor$ samples per group. Moreover, let $P_e$ denote the probability of error for a fair use hypothesis test with $\epsilon \in [-0.5, 0.5]$. The worst-case probability of error over all distributions and all decision functions $\Psi : (\mathcal{D}, h_p, h_0) \to \{0, 1\}$ satisfies*

$$\min_{\Psi} \max_{P_{\mathbf{X},\mathbf{G},Y}} P_e \geq 1 - \frac{1}{\sqrt{d}} \left(1 + 4\epsilon^2\right)^{m/2}. \tag{1}$$

*Proof.* Rather than considering all pairs of classifiers $(h_p, h_0)$ that satisfy a hypothesis, without loss of generality, we can fix $h_0$ and consider all $h_p$ that satisfies the null or alternative hypothesis.

Formally, define $V(h) := \mathbb{1}[\gamma(h, h_0) \geq \epsilon]$, which is the indicator function that $h$ does not satisfy the null hypothesis. Hence, for each classifier $h$, $V(h) = 1$ if and only if $(h, h_0) \in H_1$.

The probability of error in hypothesis test can be written as:

$$P_e = \Pr\left(\Psi(\mathcal{D}, h, h_0) = 1 \mid V(h) = 0\right) + \Pr\left(\Psi(\mathcal{D}, h, h_0) = 0 \mid V(h) = 1\right).$$

The minimax probability of error can be bounded as:

$$\begin{aligned}
\min_{\substack{\Psi \\ P_0 \in H_0 \\ P_1 \in H_1}} \max P_e &= \min_{\substack{\Psi \\ P_0 \in H_0 \\ P_1 \in H_1}} \max \Pr\left(\Psi(h) = 1 \mid V(h) = 0\right) + \Pr\left(\Psi(h) = 0 \mid V(h) = 1\right) \\
&\geq \max_{\substack{P_0 \in H_0 \\ P_1 \in H_1}} \min_{\Psi} \mathrm{P}_0\left(\Psi(h) = 1\right) + \mathrm{P}_1\left(\Psi(h) = 0\right) \\
&= \max_{\substack{P_0 \in H_0 \\ P_1 \in H_1}} \min_{\Psi} 1 - \mathrm{P}_0\left(\Psi(h) = 0\right) + \mathrm{P}_1\left(\Psi(h) = 0\right) \\
&\geq \max_{\substack{P_0 \in H_0 \\ P_1 \in H_1}} 1 - \sup_{A \text{ measurable}} \left(\mathrm{P}_0(A) - \mathrm{P}_1(A)\right) \\
&= \max_{\substack{P_0 \in H_0 \\ P_1 \in H_1}} 1 - \mathsf{TV}(P_0 || P_1).
\end{aligned}$$

$P \in H_0$ and $Q \in H_1$ can be chosen to lower bound the minimax probability of error by

$$\begin{aligned}
\min_{\substack{\Psi \\ P_0 \in H_0 \\ P_1 \in H_1}} \max P_e &\geq \max_{\substack{P_0 \in H_0 \\ P_1 \in H_1}} 1 - \mathsf{TV}(P_0 || P_1) \\
&\geq 1 - \mathsf{TV}(P || Q). \tag{2}
\end{aligned}$$

Let $\zeta_j = (\zeta_j^1, ..., \zeta_j^m) \in \{-1, +1\}^m$, and define $P$ and $Q$ as

$$P(\zeta_1, ..., \zeta_d) \triangleq \frac{1}{d} \sum_{i=1}^{d} P_i^*(\zeta_i) \prod_{j \neq i} P_j(\zeta_j), \tag{3}$$

$$Q(\zeta_1, ..., \zeta_d) \triangleq \prod_{k=1}^{d} P_j(\zeta_j), \tag{4}$$

where

$$P_i(\zeta_i) \triangleq \prod_{k=1}^{m} \Pr(\mathsf{Cat}_{\frac{1}{2}, 0, \frac{1}{2}} = \zeta_i^k),$$

$$P_i^*(\zeta_i) \triangleq \prod_{k=1}^m \Pr(\mathsf{Cat}_{\frac{1}{2}+\epsilon,0,\frac{1}{2}-\epsilon} = \zeta_i^k).$$

Where $\mathsf{Cat}_{p_1,1-p_1-p_2,p_2}$ is the ternary categorical distribution, i.e., $\mathsf{Cat}_{p_1,1-p_1-p_2,p_2} = -1$ with probability $p_1$, $\mathsf{Cat}_{p_1,1-p_1-p_2,p_2} = 1$ with probability $p_2$, and $\mathsf{Cat}_{p_1,1-p_1-p_2,p_2} = 0$ with probability $1 - p_1 - p_2$.

Given $P$ and $Q$ we can find an upper bound for the total variation is given by:

$$\mathsf{TV}\,(P||Q) = \frac{1}{2} \sum_{\zeta_1,...,\zeta_d} |P(\zeta_1,...,\zeta_d) - Q(\zeta_1,...,\zeta_d)| \tag{5}$$

$$= \frac{1}{2} \sum_{\zeta_1,...,\zeta_d} \left| \frac{1}{d} \sum_{i=1}^d P_i^*(\zeta_i) \prod_{j \neq i} P_j(\zeta_j) - \prod_{k=1}^d P_j(\zeta_j) \right| \tag{6}$$

$$= \frac{1}{2} \sum_{\zeta_1,...,\zeta_d} \left| \frac{1}{d} \sum_{i=1}^d \frac{P_i^*(\zeta_i)}{P_i(\zeta_i)} \prod_{k=1}^d P_j(\zeta_j) - \prod_{k=1}^d P_j(\zeta_j) \right| \tag{7}$$

$$= \frac{1}{2} \mathbb{E}\left[ \left| \frac{1}{d} \sum_{i=1}^d \frac{P_i^*(\zeta_i)}{P_i(\zeta_i)} - 1 \right| \right] \tag{8}$$

$$= \frac{1}{2} \mathbb{E}\left[ \left| \frac{1}{d} \sum_{i=1}^d \prod_{k=1}^m (1+2\epsilon)^{\frac{1-\zeta_i^k}{2}} (1-2\epsilon)^{\frac{1+\zeta_i^k}{2}} - 1 \right| \right] \tag{9}$$

Define $\hat{\zeta}_i \triangleq (1-\zeta_i)/2$ (element wise). Hence, each entry of $z_i$ is distributed as a Bernoulli distribution with parameter $1/2$.

$$\mathsf{TV}\,(P||Q) = \frac{1}{2} \mathbb{E}\left[ \left| \frac{1}{d} \sum_{i=1}^d \prod_{k=1}^m (1+2\epsilon)^{\frac{1-\zeta_i^k}{2}} (1-2\epsilon)^{\frac{1+\zeta_i^k}{2}} - 1 \right| \right]$$

$$= \frac{1}{2} \mathbb{E}\left[ \left| \frac{1}{d} \sum_{i=1}^d \prod_{k=1}^m (1+2\epsilon)^{\hat{\zeta}^k} (1-2\epsilon)^{1-\hat{\zeta}^k} - 1 \right| \right]$$

$$= \frac{1}{2} \mathbb{E}\left[ \left| \frac{1}{d} \sum_{i=1}^d (1+2\epsilon)^{\sum_{k=1}^m \hat{\zeta}^k} (1-2\epsilon)^{m-\sum_{k=1}^m \hat{\zeta}^k} - 1 \right| \right]$$

Define $z_i \triangleq \sum_{k=1}^m \hat{\zeta}_i^k$. Then, $z_i$ is distributed as a Binomial distribution with parameter $1/2$.

$$\mathsf{TV}\,(P||Q) = \frac{1}{2} \mathbb{E}\left[ \left| \frac{1}{d} \sum_{i=1}^d (1+2\epsilon)^{\sum_{k=1}^m \hat{\zeta}^k} (1-2\epsilon)^{m-\sum_{k=1}^m \hat{\zeta}^k} - 1 \right| \right]$$

$$= \frac{1}{2} \mathbb{E}\left[ \left| \frac{1}{d} \sum_{i=1}^d (1+2\epsilon)^{z_i} (1-2\epsilon)^{m-z_i} - 1 \right| \right]$$

$$\leq \mathbb{E}\left[ \left| \frac{1}{d} \sum_{i=1}^d (1+2\epsilon)^{z_i} (1-2\epsilon)^{m-z_i} - 1 \right|^2 \right]^{1/2}, \quad \text{(by Cauchy-Schwarz)}$$

$$\leq \mathbb{E}\left[ \left( \frac{1}{d} \sum_{i=1}^d (1+2\epsilon)^{z_i} (1-2\epsilon)^{m-z_i} \right)^2 - 1 \right]^{1/2}$$

$$= \mathbb{E}\left[ \left( \frac{1}{d^2} \sum_{i,j=1}^d (1+2\epsilon)^{z_i} (1-2\epsilon)^{m-z_i} (1+2\epsilon)^{z_j} (1-2\epsilon)^{m-z_j} \right) - 1 \right]^{1/2}$$

$$= \mathbb{E}\left[ \frac{1}{d^2} \sum_{i=1}^d \left( (1+2\epsilon)^{z_i} (1-2\epsilon)^{m-z_i} \right)^2 + \left( \frac{1}{d^2} \sum_{\substack{i,j=1 \\ i \neq j}}^d (1+2\epsilon)^{z_i} (1-2\epsilon)^{m-z_i} (1+2\epsilon)^{z_j} (1-2\epsilon)^{m-z_j} \right) - 1 \right]^{1/2}$$

$$= \mathbb{E}\left[\frac{1}{d^2}\sum_{i=1}^{d}\left((1+2\epsilon)^{z_i}(1-2\epsilon)^{m-z_i}\right)^2 + \frac{d(d-1)}{d^2} - 1\right]^{1/2}$$

$$\leq \frac{1}{\sqrt{d}}\mathbb{E}\left[|(1+2\epsilon)^{z_1}(1-2\epsilon)^{m-z_1}|^2\right]^{1/2}$$

$$= \frac{1}{\sqrt{d}}(1-2\epsilon)^m\mathbb{E}\left[|(1+2\epsilon)^{z_1}(1-2\epsilon)^{-z_1}|^2\right]^{1/2}$$

$$= \frac{1}{\sqrt{d}}(1-2\epsilon)^m\left(M_{\mathsf{Bin}(1/2,m)}\left(2\log\left(\frac{1+2\epsilon}{1-2\epsilon}\right)\right)\right)^{1/2}$$

$$= \frac{1}{\sqrt{d}}(1+4\epsilon^2)^{m/2} \tag{10}$$

Consequently, combining (2) and (10)

$$\min_{\Psi}\max_{\substack{P_0\in H_0 \\ P_1\in H_1}} P_e \geq 1 - \mathsf{TV}(P\|Q)$$

$$\Rightarrow \min_{\Psi}\max_{\substack{P_0\in H_0 \\ P_1\in H_1}} P_e \geq 1 - \frac{1}{\sqrt{d}}(1+4\epsilon^2)^{m/2}.$$

$\square$

**Observation.** For the sake of clarity of the proof we flip the hypothesis test in the theorem statement. It is clear that an analogous proof holds for the case of non-flipped hypothesis test see Appendix C.4. Intuitively, estimating $\gamma$ is as hard as estimating $-\gamma$. Hence, by applying this result for $-\gamma$ we get the theorem statement in the main paper.

**Direct Extension to False Positive Rate/False Negative Rate** We can replace the accuracy metric in BoP with FPR or FNR, and the proof still holds. Our proof relies on the observation that the accuracy (of a model on a data point) can be written as a Bernoulli random variable where 1 denotes an accurate prediction and 0 otherwise. BoP is the min difference in accuracy of the two models over all groups, which can be modelled using the difference of two bernoullis. Then, we can characterize the two hypotheses by the means of the resulting Categorical distributions.

Analogously, FP and FN of a model on data points can also be represented by a Bernoulli where 1 denotes an FP/FN and 0 denotes a TP/TN. The parameters would refect the different definition, but the proof then carries out directly.

### B.2 Proof of Corollary 1

**Corollary 1** (Limit for Testing). *With the same setup as in Theorem 1. Take $n = 8\times 10^9$, and let $\epsilon = 0.01$. If $\min\max P_e \leq 1/2$ then $k \leq 19$.*

*Proof.* If $\min_{\Psi}\max_{\mathcal{D}} P_e \leq 1/2$ then:

$$1 - \frac{1}{\sqrt{d}}(1+4\epsilon^2)^{m/2} \leq \min_{\Psi}\max_{\substack{P_0\in H_0 \\ P_1\in H_1}} P_e \leq 1/2.$$

Given $d = 2^k$, $m = n/d$, $n = 8\times 10^9$, and $\epsilon = 0.01$, then

$$\phi(k) = 1 - \frac{1}{2^{k/2}}(1+0.0004)^{\frac{8\times 10^9}{2^{k+1}}} \leq 1/2.$$

Then $\phi$ is a increasing function in $k$, and $\phi(18) \approx 0.12 < 1/2$, $\phi(19) \approx 0.97 > 1/2$. Hence, there exists a phase transition in the minimax error probability between $k = 18$ and $k = 19$, i.e., for $k \leq 18$ $\min_{\Psi}\max P_e \leq 1/5$ and, for only one more group attribute ($k \geq 19$) $\min_{\Psi}\max P_e \approx 1$.

$\square$

## B.3 Proof of Lemma 1

The original Le Cam Lemma (Le Cam, 1973) states that (11) holds when $\theta_1$, $\theta_2$ are elements of $\Theta$. We generalize this lemma for the case where $\theta_1$, $\theta_2$ are elements of $\Theta$ chosen at random.

**Lemma 1** (Generalized Le Cam). *Let $\{\Lambda, d\}$ be a metric space, and $\mathcal{D}$ be a dataset. Let $\mathcal{P}$ be a family of probability distributions with parameter $\theta \in \Theta$. Suppose that $f$ is a statistics with image in $\Lambda$ ( $f(\theta)$ for $\theta \in \Theta$ means the statistics calculated for $P_\theta$), and that there exists $\Lambda_1$, $\Lambda_2$ subsets of $\Lambda$ that are $2\delta$-separated. Also, suppose that $\Theta_1$ and $\Theta_2$ are subsets of $\Theta$ such that $f(\theta) \in \Lambda_1$ for all $\theta \in \Theta_1$ and $f(\theta) \in \Lambda_2$ for all $\theta \in \Lambda_2$. Then we have that:*

$$\min_{f^* \in \sigma(\mathcal{D})} \max_{\theta \in \Theta} \mathbb{E}\left[d(f(\theta), f^*)\right] \geq \delta(1 - \mathsf{TV}(P_{\theta^1} || P_{\theta^2})), \tag{11}$$

*where $\theta^i \in \sigma(\Lambda)$ with image in $\Theta_i$, and $P_{\theta^i}$ indicates a probability distribution in $\mathcal{P}$ with parameter $\theta^i$ for $i \in [2]$.*

*Proof.* First, we write the minimax error in the set $\Theta$ as a minimax in the $\sigma$-algebra generated by $\Theta$ ($\sigma(\Theta)$). To do that, take $\pi \in \sigma(\Theta)$ and $\theta \sim \pi$. The mean is smaller or equal to the maximum, if $f^* \in \sigma(\Theta)$ then the following inequality holds:

$$\max_{\theta \in \Theta} \mathbb{E}\left[d(f(\theta), f^*)\right] = \max_{P_\theta \in \mathcal{P}} \mathbb{E}\left[d(f(\theta), f^*)\right] \geq \mathbb{E}_{P_\theta; \theta \sim \pi}\left[d(f(\theta), f^*)\right]. \tag{12}$$

Eq. (12) holds for all $\pi \in \sigma(\Theta)$, then:

$$\max_{\theta \in \Theta} \mathbb{E}\left[d(f(\theta), \theta^*)\right] \geq \max_{\pi \in \sigma(\Theta)} \mathbb{E}_{P_\theta}\left[d(f(\theta), f^*)\right] = \max_{P_\theta; \theta \in \sigma(\Theta), P \in \mathcal{P}} \mathbb{E}_{P_\theta; \theta \sim \pi}\left[d(f(\theta), f^*)\right] \tag{13}$$

By using the Le Cam Lemma in Le Cam (1973) with the statistic being $f$, $\theta^i \in \sigma(\Theta)$ with image in $\Theta_i$, the metric space being $\{\Lambda, d\}$, and the $2\delta$-separated sets being $\Lambda_1$, $\Lambda_2$ where $f(P_\theta^i) \in \Lambda_i$ for all $i \in [2]$, we conclude that:.

$$\max_{\theta \in \Theta} \mathbb{E}\left[d(f(\theta), \theta^*)\right] \geq \max_{P_\theta; \theta \in \sigma(\Theta), P \in \mathcal{P}} \mathbb{E}_{P_\theta; \theta \sim \pi}\left[d(f(\theta), f^*)\right] \geq \delta(1 - \mathsf{TV}(P_{\theta^1} || P_{\theta^2})) \tag{14}$$

Hence, combining (12), (13), and (14) together, then

$$\min_{\theta^* \in \sigma(\mathcal{D})} \max_{\theta \in \Theta} \mathbb{E}\left[d(f(\theta), \theta^*)\right] \geq \min_{\theta^* \in \sigma(\mathcal{D})} \max_{\pi \in \sigma(\Theta)} \mathbb{E}_{\theta \sim \pi}\left[d(f(\theta), \theta^*)\right] \geq \delta(1 - \mathsf{TV}(P_{\theta^1} || P_{\theta^2})) \tag{15}$$

$\square$

## B.4 Proof of Theorem 2

**Theorem 2** (Minimax bounds for the Estimation MSE). *Given a dataset $\mathcal{D}$, constants $d, m \in \mathbb{N}$ such that $d \geq 4$, $d \leq 2^{m+2}$, a group-blind, and a personalized classifier, the minimax estimation error for the benefit of personalization in (2) is bounded as*

$$\frac{\log(d)}{16m} - \frac{\log(4)}{16m} \leq \inf_{\gamma^* \in \sigma(\mathcal{D})} \sup_{P_{\mathbf{X}, \mathbf{G}, Y}} \mathbb{E}\left[(\gamma - \gamma^*)^2\right] \leq \frac{\log(d)}{m} + \frac{\log(m)}{m} + \frac{2 + \log(2)}{m}, \tag{16}$$

*Proof.* **Lower Bound.** From Definition 2 the BoP is given by:

$$\gamma(h_0, h_g, P_\mathcal{D}) = \min_{\mathbf{g} \in \mathcal{G}} P_j(Y = h_g(X, S)) - P_j(Y = h_0(X))$$
$$= \min_{\mathbf{g} \in \mathcal{G}} \Pr(Y = h_g(X, S) \mid S = j) - \Pr(Y = h_0(X) \mid S = j).$$

Define $\theta_j^g \triangleq \Pr(Y = h_g(X, S) \mid S = j)$ and $\theta_j^0 \triangleq \Pr(Y = h_0(X) \mid S = j)$. Let $\Theta \triangleq (\Theta^g, \Theta^0) \triangleq (\theta_1^g, ..., \theta_d^g, \theta_1^0, ..., \theta_d^0)$, and define $\gamma(\Theta) \triangleq \min_{\mathbf{g} \in \mathcal{G}} \Pr(\mathsf{Ber}(\theta_0^g) = 1) - \Pr(\mathsf{Ber}(\theta_j^0) = 1)$. Then $\gamma(h_0, h_g, P_\mathcal{D}) = \gamma(\Theta)$.

Hence:

$$\inf_{\gamma^* \in \sigma(\mathcal{D})} \sup_{P_{\mathbf{X},\mathbf{G},Y}} \mathbb{E}\left[(\gamma - \gamma^*)^2\right] = \inf_{\gamma^* \in \sigma(\mathcal{D})} \sup_{\substack{\Theta^g \in [0,1]^d \\ \Theta^0 \in [0,1]^d}} \mathbb{E}\left[(\gamma(\Theta) - \gamma^*)^2\right]$$

By Lemma 1 we can lower bound the minimax of the MSE by:

$$\inf_{\gamma^* \in \sigma(\mathcal{D})} \sup_{\substack{\Theta^g \in [0,1]^d \\ \Theta^0 \in [0,1]^d}} \mathbb{E}\left[(\gamma(\Theta) - \gamma^*)^2\right] \geq \frac{||\gamma(\theta^1) - \gamma(\theta^2)||_2^2}{2}(1 - \mathsf{TV}(P_{\gamma(\theta^1)}||P_{\gamma(\theta^2)}))$$

$$\geq \frac{||\gamma(\theta^1) - \gamma(\theta^2)||_2^2}{2}(1 - \mathsf{TV}(P_{\theta^1}||P_{\theta^2}).$$

Where the last inequality comes from the data processing inequality.

Note that, as in Theorem 1, we can take a categorical distribution mixture. This follows from the fact that given $C^p \sim \mathsf{Ber}(\theta_j^g)$, $C^0 \sim \mathsf{Ber}(\theta_j^0)$, and $C_j = C^p - C^0 \sim \mathsf{Cat}(\theta_{-1,j}, \theta_{1,j})$ then $\theta_j^g - \theta_j^0 = \theta_{1,j} - \theta_{-1,j}$. Hence, the BoP can be rewritten as $\min_j \theta_{1,j} - \theta_{-1,j}$.

By the proof of Theorem 1, and the above observation, we can choose $P_{\theta^1}$ and $P_{\theta^2}$ as in (3) and (4), respectively. Hence, we have that:

- $||\gamma(\theta^1) - \gamma(\theta^2)||_2^2 = \epsilon^2$
- $(1 - \mathsf{TV}(P_{\theta^1}||P_{\theta^2}) \geq 1 - \frac{1}{\sqrt{d}}(1 + 4\epsilon^2)^{m/2}$

Then:

$$\inf_{\gamma^* \in \sigma(\mathcal{D})} \sup_{P_{\mathbf{X},\mathbf{G},Y}} \mathbb{E}\left[(\gamma - \gamma^*)^2\right] \geq \frac{\epsilon^2}{2}(1 - \frac{1}{\sqrt{d}}(1 + 4\epsilon^2)^{m/2}). \tag{17}$$

Take $\epsilon^2 = \left((d/4)^{1/m} - 1\right)/4$. Note that, once $4 < d < 2^{m+2}$ then $0 < \epsilon^2 < 1/4$ and $\epsilon < 1/2$. Hence, by plugging in $\epsilon^2$:

$$\inf_{\gamma^* \in \sigma(\mathcal{D})} \sup_{P_{\mathbf{X},\mathbf{G},Y}} \mathbb{E}\left[(\gamma - \gamma^*)^2\right] \geq \frac{\left((d/4)^{1/m} - 1\right)}{16}$$

$$\geq \frac{\log\left((d/4)^{1/m}\right)}{16}$$

$$= \frac{\log(d)}{16m} - \frac{\log(4)}{16m}$$

Similar to Theorem 1, the proof of the lower bound using FPR and FNR as benchmark metrics follows directly.

**Upper Bound.** Let the empirical estimator be $\hat{\gamma}$ to be the $\min_j \theta_j^{*(g)} - \theta_j^{*(0)}$ where:

$$\theta_j^{*(g)} = \frac{\sum_{i=1}^m \mathbb{1}_{h_g(x_i, j) = y_i | s_i = j}}{m}$$

$$\theta_j^{*(0)} = \frac{\sum_{i=1}^m \mathbb{1}_{h_0(x_i) = y_i | s_i = j}}{m}$$

Then:

$$\inf_{\gamma^*} \sup_{\Theta \in [0,1]^{2d}} \mathbb{E}\left[(\gamma(\Theta) - \gamma^*)^2\right] \leq \sup_{\Theta \in [0,1]^{2d}} \mathbb{E}\left[(\gamma(\Theta) - \hat{\gamma})^2\right]$$

$$= \sup_{\Theta \in [0,1]^{2d}} \mathbb{E}\left[(\min_{\mathbf{g} \in \mathcal{G}}(\Theta_j^{(p)} - \Theta_j^{(0)}) - \min_{\mathbf{g} \in \mathcal{G}}(\theta_j^{*(p)} - \theta_j^{*(0)}))^2\right] \tag{18}$$

From the fact the if $a_j, b_j \in \mathbb{R}$ then $|\min_j a_j - \min_j b_j| \leq \max_j |a_j - b_j|$, the term in (18) is upper bounded by

$$\leq \sup_{\Theta \in [0,1]^{2d}} \mathbb{E}\left[ (\max_{\mathbf{g} \in \mathcal{G}} |\Theta_j^{(p)} - \Theta_j^{(0)} - \theta_j^{*(p)} + \theta_j^{*(0)}|)^2 \right]$$

$$= \sup_{\Theta \in [0,1]^{2d}} \mathbb{E}\left[ \max_{\mathbf{g} \in \mathcal{G}} (|\Theta_j^{(p)} - \Theta_j^{(0)} - \theta_j^{*(p)} + \theta_j^{*(0)}|)^2 \right]$$

$$\leq \sup_{\Theta \in [0,1]^{2d}} 2\mathbb{E}\left[ \max_{\mathbf{g} \in \mathcal{G}} |\Theta_j^{(p)} - \theta_j^{*(p)}|^2 \right] + 2\mathbb{E}\left[ \max_{\mathbf{g} \in \mathcal{G}} |\Theta_j^{(0)} - \theta_j^{*(0)}|^2 \right]$$

Then the minimax estimation error for the benefit of personalization is upper bounded by:

$$\inf_{\gamma^*} \sup_{\Theta \in [0,1]^{2d}} \mathbb{E}\left[ (\gamma(\Theta) - \gamma^*)^2 \right] \leq \sup_{\Theta \in [0,1]^{2d}} 2\mathbb{E}\left[ \max_{\mathbf{g} \in \mathcal{G}} |\Theta_j^{(p)} - \theta_j^{*(p)}|^2 \right] + 2\mathbb{E}\left[ \max_{\mathbf{g} \in \mathcal{G}} |\Theta_j^{(g)} - \theta_j^{*(g)}|^2 \right]$$

$$\leq 4 \sup_{\Theta \in [0,1]^d} \mathbb{E}\left[ \max_{\mathbf{g} \in \mathcal{G}} |\Theta_j - \theta_j^{*(g)}|^2 \right] \qquad (19)$$

It is necessary to upper bound $\mathbb{E}\left[ \max_{\mathbf{g} \in \mathcal{G}} |\Theta_j - \theta_j^*|^2 \right]$. This can be done by using conditioning and a concentration result:

$$\mathbb{E}\left[ \max_{\mathbf{g} \in \mathcal{G}} |\Theta_j - \theta_j^*|^2 \right] = \mathbb{E}\left[ \max_{\mathbf{g} \in \mathcal{G}} |\Theta_j - \theta_j^*|^2 \Big| \max_{\mathbf{g} \in \mathcal{G}} |\Theta_j - \theta_j^*|^2 \geq \epsilon \right] \Pr\left( \max_{\mathbf{g} \in \mathcal{G}} |\Theta_j - \theta_j^*|^2 \geq \epsilon \right)$$

$$+ \mathbb{E}\left[ \max_{\mathbf{g} \in \mathcal{G}} |\Theta_j - \theta_j^*|^2 \Big| \max_{\mathbf{g} \in \mathcal{G}} |\Theta_j - \theta_j^*|^2 < \epsilon \right] \Pr\left( \max_{\mathbf{g} \in \mathcal{G}} |\Theta_j - \theta_j^*|^2 < \epsilon \right)$$

Since $\max_{\mathbf{g} \in \mathcal{G}} |\Theta_j - \theta_j^*|^2 \leq 1$, then

$$\mathbb{E}\left[ \max_{\mathbf{g} \in \mathcal{G}} |\Theta_j - \theta_j^*|^2 \right] \leq \Pr\left( \max_{\mathbf{g} \in \mathcal{G}} |\Theta_j - \theta_j^*|^2 \geq \epsilon \right) + \epsilon$$

$$\leq \epsilon + 1 - \Pr\left( \max_{\mathbf{g} \in \mathcal{G}} |\Theta_j - \theta_j^*|^2 < \epsilon \right)$$

$$\leq \epsilon + 1 - \prod_{j=1}^{d} \Pr\left( |\Theta_j - \theta_j^*|^2 < \epsilon \right)$$

$$\leq \epsilon + 1 - \prod_{j=1}^{d} (1 - \Pr\left( |\Theta_j - \theta_j^*|^2 \geq \epsilon \right))$$

$$\leq \epsilon + 1 - (1 - 2e^{-2n\epsilon})^d \qquad (20)$$

Hence, combining (19) and (20)

$$\inf_{\gamma^* \in \sigma(\mathcal{D})} \sup_{P_{\mathbf{X}, \mathbf{G}, Y}} \mathbb{E}\left[ (\gamma - \gamma^*)^2 \right] \leq 4 \left( \epsilon + 1 - (1 - 2e^{-2n\epsilon})^d \right).$$

Take $\epsilon = \frac{c \log(d)}{2m}$ and $c = 1 + \log_d(2m)$, then we conclude that:

$$\inf_{\gamma^* \in \sigma(\mathcal{D})} \sup_{P_{\mathbf{X}, \mathbf{G}, Y}} \mathbb{E}\left[ (\gamma - \gamma^*)^2 \right] \leq \frac{\log(d)}{m} + \frac{\log(m)}{m} + \frac{2 + \log(2)}{m}$$

Similarly, for the upperbound proof using FPR and FNR, the only difference is the choice of $\theta_j^*$. For FPR we define $\theta_j^*(g) = \sum \frac{\mathbb{1}_{hg(x_i, j)=1 | s_i = j, y_i = 0}}{m}$ and for FNR we choose $\theta_j^*(g) = \sum \frac{\mathbb{1}_{hg(x_i, j)=0 | s_i = j, y_i = 1}}{m}$. $\qquad \square$

## B.5    Proof of Corollary 2

**Corollary 2** (Limits of Estimation). *Given a dataset $\mathcal{D}$ with $n$ data points, $d \geq 4$ groups, $m = n/d$ data points per group, an $0 < \delta < 1/16$, a group-blind and a personalized classifier. If $d \leq 2^{m+2}$, and*

$$\inf_{\gamma^* \in \sigma(\mathcal{D})} \sup_{P_{\mathbf{X}, \mathbf{G}, Y}} \mathbb{E}\left[(\gamma - \gamma^*)^2\right] \leq \delta,$$

*then, the number of binary group attributes $(k)$ satisfies*

$$(k - 2)2^k \leq \frac{16}{\ln 2} n\delta. \tag{21}$$

*Specifically, for $n = 8 \times 10^9$ (approximately the number of humans on the Earth) and $\delta = 1/400 = (5\%)^2$, then the maximum number of binary group attributes is 25, i.e, $k \leq 25$ in order to ensure reliable estimation.*

*Proof.*  **Rule of thumb.** If $\inf_{\gamma^* \in \sigma(\mathcal{D})} \sup_{P_{\mathbf{X}, \mathbf{G}, Y}} \mathbb{E}\left[(\gamma - \gamma^*)^2\right] \leq \delta$ then:

$$\frac{\log(d)}{16m} - \frac{\log(4)}{16m} \leq \delta$$

For $k$ binary attributes and $d = 2^k$, $m = n/d$. Hence,

$$\log(d) \leq \log(4) + \delta 16m$$

$$\Rightarrow k \log(2) \leq 2 \log(2) + \delta 16 \frac{n}{2^k}$$

$$\Rightarrow 2^k k \log(2) \leq 2^{k+1} \log(2) + \delta 16n$$

Then

$$(k - 2)2^k \leq \frac{16n}{\log(2)}\delta. \tag{22}$$

**Infeasibility result.** Selecting $n = 8 \times 10^9$, and $\delta = 1/400$ in (22).

$$(k - 2)2^k \leq \frac{128 \times 10^9}{400 \log(2)}$$

Note that $(k - 2)2^k$ is an increasing function of $k$ when $k \geq 2$. Moreover, $(25 - 2)2^{25} > \frac{128 \times 10^9}{400 \log(2)}$. Hence, $k \leq 25$. $\qquad\square$

## C    Further Insights on Our Results

In this section, we offer further insights on the significance and usefulness of our main results and clarify our assumptions.

### C.1    the BoP Metric: Significance and Usefulness

The paper considers the problem of reliably estimating the BoP, regardless if it is positive (all groups benefit) or negative (at least one group is harmed).

If the number of samples allows the BoP to be reliably estimated across all groups, then the next step is to decide which groups should use a personalized model or not. However, if the BoP cannot be precisely estimated, then the decision of deploying a personalized model becomes challenging: even though we may estimate a gain in personalization for some groups, it can be (information-theoretically) infeasible to tell if that gain is statistically significant, or simply a fluke due to lack of samples or too many groups. Our bounds precisely capture the trade-off between number of groups, sample size, and gain in personalization.

Second, the BoP is designed to empower data holders, i.e. users of the ML model. A user's group attribute may not be known a priori and may incur a collection cost – particularly in healthcare applications. For example, group attribute collection can require invasive procedures (e.g., blood draw to determine HIV status) or require that the user reveal private information to the model holder

(e.g., substance use). In such cases, it is essential that the collection and disclosure of group attributes ensure a gain in accuracy, regardless of what the value of the attribute is. Alternatively, the user should be informed if there is a chance that they may not receive a prediction gain with the disclosure of a group attribute (i.e., non-positive BoP). This information can enable the user to make an informed decision to collect and disclose personal data (e.g., if a blood draw merits the gain in diagnostics accruacy by a ML model). Either way, if the value of a group attribute is not known a priori, we must reliably estimate the BoP for all groups – this is exactly what our bounds capture.

## C.2 Minimax Bounds: Significance and Usefulness

We bound the minimax errors of testing and estimating the BoP in Theorem 1 and 2, respectively. We offer more insights on what minimax errors mean and the significance and usefulness of bounding minimax errors.

**Significance.** It is important to control errors in the minimax sense because the underlying dataset distribution (defined in Section 2.1) is unknown. By considering minimax bounds, we take a conservative stance to bound errors for the worst-case scenario.

In particular, in Section 3, we consider the problem of testing whether there is high enough Benefit of Personalization, and, in Theorem 1, characterize the lower bound of the minimax probability of error $P_e$ as follows:

$$\min_{\Psi} \max_{\substack{P_0 \in H_0 \\ P_1 \in H_1}} P_e \geq 1 - \frac{1}{\sqrt{d}} \left(1 + 4\epsilon^2\right)^{m/2} \tag{23}$$

In the inner layer of the right-hand side of the inequality above, we are maximizing $P_e$ over the space of distributions of $(X, Y, S)$ as defined in Section 2.1. Effectively, we are picking the data distribution $\mathcal{D}$ that maximizes the probability of error $P_e$ of distinguishing between the two hypotheses. In the outer layer, as to how empirical risk minimization goes, we choose the decision function $\Psi$ that minimizes the error.

In section 4, we consider the problem of reliable estimation and characterize the minimax MSE as:

$$\frac{\log(d)}{16m} - \frac{\log(4)}{16m} \leq \inf_{\gamma^* \in \sigma(\mathcal{D})} \sup_{P_{\mathbf{X}, \mathbf{G}, Y}} \mathbb{E}\left[(\gamma - \gamma^*)^2\right] \leq \frac{\log(d)}{m} + \frac{\log(m)}{m} + \frac{2 + \log(2)}{m}, \tag{24}$$

The minimax bounds can be analyzed analogously. In the inner layer, we are maximizing the MSE over the space of dataset distributions $\mathcal{D}$. In the outer layer, we choose a BoP estimator $\gamma^*$ that minimizes the MSE for this worst-case distribution.

In both theorems, the minimax analysis bounds errors under the worst-case distribution. Without prior knowledge of the data distribution, we cannot rule out this worst-case scenario. Thus, the significance of our bounds is that it holds for all decision functions and estimators and inherently consider the worst-case distribution for each case.

**Usefulness.** The minimax MSE upper bound in (24) always holds. Since the upper bound holds when we take the sup over dataset distributions the bound works for all distributions. Indeed, as we can see in Figure 4 and Figure 5, the MSE estimations always fall below the upper bounds.

In addition, notice that the upper bound is tight in the sense that it has the same order as the lower bound—both bounds are of the order of $O\left(\frac{\log(d \times m)}{m}\right)$. The ratios of the bounds and $\frac{\log(d \times m)}{m}$ is bounded by a constant for large enough $d$ and $m$.

In contrast, the minimax MSE lower bound in (24) does not always hold in experiments, as we can see in Fig. 4 and Fig. 5. This is, of course, expected. Minimax bounds are conservative: they bound the MSE of a worst-case distribution $P_{\mathbf{X}, \mathbf{G}, Y}$ that achieves the sup MSE under the best possible estimator. The datasets in our experiments are not quite the worst-case distribution and, hence, result in a more optimistic MSE. Nevertheless, with no prior knowledge of the dataset, the minimax lower bound is almost the best we can achieve. Surprisingly, it does seem that the distributions derived from real-world datasets in our experiments are not far from the worst-case one in terms of MSE.

Furthermore, the minimax bounds guide the maximum number of group attributes we can have in data collection. In Corollary 1 and 2, we show that to ensure reliable testing and estimation of BoP, we cannot have more than 19 and 25 features, respectively.

### C.3 Clarification on Assumptions

**Same number of data points per group.** In the setup (Section 2.1), we assume an equal number of data points per group to simplify analysis. This is the best-case scenario, and otherwise, the group with the fewest samples will dominate testing or estimation performance.

In the case of an unequal number of samples per group, the number of features we can have for reliable testing and estimation will only shrink. Since the total number of data points are fixed, an unequal number of samples per group means that some group has fewer samples. Consider a specific setup in the proof of minimax lower bounds (Section B.1, B.4). The lower bound is obtained by considering a pair of distributions $P$ and $Q$ that are difficult to distinguish. Specifically, no group in $Q$ is harmed, and only one group, chosen randomly, is harmed in $P$. In testing, differentiating between the two hypotheses boils down to identifying the one group that has negative BoP. Analogously in estimation, we need to find the group with the smallest gain of personalization and use the points in the group to estimate the BoP. Suppose the group that is harmed is the one with fewer samples, both the MSE and the probability of error will be larger.

**Binary group attributes.** We also assume in the setup (Section 2.1) that the group attributes are binary. If a group attribute has more than two categories (e.g., race), we can simply take the one-hot encoding of the attribute.

### C.4 Alternative Formulation of Hypothesis Test

In Section 3.1, we consider the hypothesis test with a positive threshold $\epsilon$:

- $H_0$: $\gamma < 0 \Leftrightarrow \min_{\mathbf{g} \in \mathcal{G}} P_j(Y = h_g(X, S)) - P_j(Y = h_0(X)) < 0$

- $H_1$: $\gamma \geq \epsilon \Leftrightarrow \min_{\mathbf{g} \in \mathcal{G}} P_j(Y = h_g(X, S)) - P_j(Y = h_0(X)) \geq \epsilon$

Alternatively, we can consider the formulation with a negative threshold:

- $H_0$: $\gamma < \epsilon \Leftrightarrow \min_{\mathbf{g} \in \mathcal{G}} P_j(Y = h_g(X, S)) - P_j(Y = h_0(X)) \leq \epsilon$

- $H_1$: $\gamma \geq 0 \Leftrightarrow \min_{\mathbf{g} \in \mathcal{G}} P_j(Y = h_g(X, S)) - P_j(Y = h_0(X)) > 0$

The alternative formulation takes a pessimistic stance. The null hypothesis states that the BoP is worse than a negative threshold, while the alternative hypothesis takes BoP to be positive, equivalent to the beneficence condition in fair use. The minimax bounds do not change with the different formulations. As observed at the end of Section B.1, estimating $\gamma$ and $-\gamma$ are analogous.

Hypothesis tests with a negative threshold display phase transition prominently when the true BoP is negative. As shown in the heat maps (Figure 2 and Figure 3), phase transition, i.e., drastic rise in probability of error, occurs when the threshold drops below -0.1 in both HSLS and COMPAS dataset. From Table 3, the estimated true BoP is -1.1 and -0.9, respectively, which is why we observe phase transition around a threshold of -0.1.

## D    Additional Experiments & Discussions

This section provides details on the experiments, explains how to reproduce the plots, and discusses how the plots illustrate our theoretical results. We conduct experiments to test and estimate the Benefit of Personalization (BoP) on three datasets: Adult (in the main paper), High School Longitudinal Study of 2009 (HSLS), and COMPAS. Our implementation (attached to the Supplementary Material with datasets) serve as an example of how to test and estimate BoP in practice.

**Table 2:** Datasets, group features, and prediction task.

| Dataset | Group Features | Prediction Task |
|---------|----------------|-----------------|
| Adult | working class, education, marital status, occupation, relationship, race, sex, and country. | Is income higher than $50k |
| HSLS | sex, race, parent's current employment status, and family income (in categories). | Is math grade higher than the median |
| COMPAS | race and gender. | Recidivism in two years |

### D.1  Dataset

We use three datasets (Adult, HSLS, and COMPAS) for all the experiments in this paper. For each dataset, we separate the features into two classes, group features and non-group features. We also define for each dataset the label to be predicted. This information is summarized in Table 2.

Because the number of data points per group varies with the group, to approximate the actual dataset distribution, we sample with replacement from it to create new datasets with an equal number of data points per group. We call this new data set a *Semi-Synthetic* dataset. Next, we discuss in more detail how the semi-synthetic data was generated.

**Semi-Synthetic data.**   For each dataset, we select all intersectional groups (e.g., single women from South America working in a c-level job) with more than a certain number of data points – we call that *significant groups*. After that, we choose $m$, the number of data points per group. Then, we sample $m$ data points with replacement for each significant group. Finally, we define the semi-synthetic dataset as the dataset generated by the collection of all $m$ data points for all significant groups.

More specifically, for HSLS, significant groups have more than 50 data points, generating $d = 87$ significant groups. For COMPAS, significant groups have more than 20 data points, generating $d = 9$ significant groups. We only use the semi-synthetic dataset for testing; we train models in the original train dataset.

### D.2  Classifiers

For all three datasets, we train personalized and group blind classifiers. We use the same model for the group blind and personalized; the only difference is that the personalized classifier uses the group attributes described in Table 2, while the group bling does not.

For the COMPAS datasets, we train classifiers using logistic regression. For HSLS, we train classifiers using random forest. For all classifiers, we use the default hyperparameters from Scikitlearn v1.0.2 Pedregosa et al. (2011). Also, for training the classifiers, we split the dataset into $50\%$ for training and $50\%$ for the test. Finally, we use the test dataset to produce semi-synthetic datasets and calculate the probability of error and the MSE.

### D.3  Estimation and Test

**Benefit of Personalization.**   Since we do not have access to the dataset distribution, we define the ground truth BoP as the BoP given by the semi-synthetic testing dataset with 1000 samples per group. The values of BoP for all models are given in Table 3. For all models, the BoP is negative, so there exists a group being harmed. Indeed, for the HSLS dataset, this harm is $11\%$, i.e., there exists a group where the group blind classifier performs way better ($11\%$) than the personalized one.

**Testing.**   For the hypothesis test, we use a threshold test featured in Section 3 and Appendix C. In Figures 2, and 3 **Left** we show the probability of error in a hypothesis test (threshold test). For the HSLS, we detect a phase transition on the error probability when the number of data points per group is equal to the number of groups. For COMPAS, the phase transition happens when the number of data points per group equals 20. When the number of data points per group is equal to the number of

**Table 3:** Estimation of the BoP and the corresponding subgroup that produces the BoP using semi-synthetic dataset.

| Dataset | Benefit of Personalization | Group Most Harmed |
|---------|---------------------------|-------------------|
| Adult | -10% | ('Private', 'HS-grad', 'Married-civ-spouse', 'Tech-support', 'Husband', 'White', 'Male', 'United-States') |
| HSLS | -11% | ('X1SEX':'1.0', 'X1RACE':'2.0', 'P1JOBNOW1':'1.0', 'X1FAMINCOME':'3.0') |
| COMPAS | -9% | 'Asian', 'Male') |

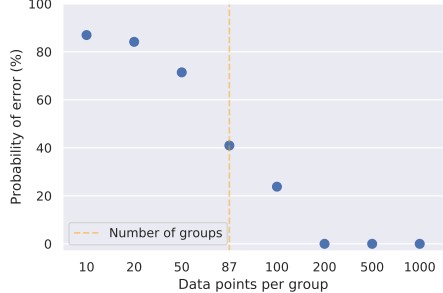 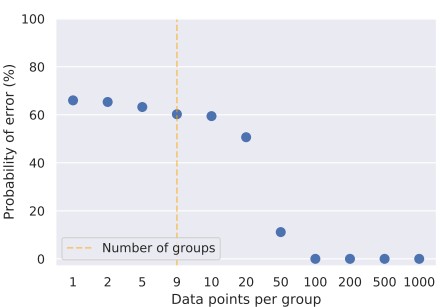

**Figure 1:** Maximum of 0 and lower bound in (1) for $\epsilon = 0.1$ is in the y-axis, and $m$ is in the x-axis. The orange line represents when the number of data points per group is equal to the number of groups. **Left:** This shows the minimax probability of error for the parameters of the **HSLS** dataset, i.e., $(d = 87)$. **Right:** his shows the minimax probability of error for the parameters of the **Compas** dataset, i.e., $(d = 9)$.

groups, the probability of error is more than $60\%$. Then, in both cases, the practitioner should, at the very minimum, ensure that the number of points per group is bigger than the number of groups.

**Phase transition.**   In Figure 1, we plot the minimax error probability in Theorem 1 for HSLS **Left** and COMPAS **Right**. Note that the minimax probability of error predicts when the probability of error is higher than $50\%$. Figure 1 **Left** shows that for the HSLS dataset, when the number of data points per group is smaller than $87$ the probability of error is bigger then $50\%$ that agrees with the numerical experiment in Figure 2 **Left**. Figure 1 **Right** shows that for the COMPAS dataset, when the number of data points per group is smaller than $20$ the probability of error is greater than $50\%$ and agrees with the numerical experiment in Figure 3 **Left**.

We also show in Figures 2, and 3 **Right** a heat map showing how the probability of error in hypothesis test behaves for different values of threshold and data points per group. From those figures, we conclude that when the threshold is way bigger than the true BoP value, the probability of error decreases, adding more "tolerance" to the test.

**Estimation.**   Throughout this paper, we use the empirical estimator for the BoP, i.e., for each group $\mathbf{g} \in \mathcal{G}$ we calculate the empirical probability of $h_g(X, j) = Y$ and $h_0(X) = Y$. We define the BoP as the minimum difference of those estimated values.

The behavior of our bounds for the minimax MSE is illustrated in Fig. 4, and 5. Again, as seen in the experiments using the Adult dataset, even knowing that our bounds are minimax, for a given distribution – HSLS and COMPAS dataset distribution – they are also sharp. Hence, they can – and should – be used to guide a practitioner on how many groups features to collect.

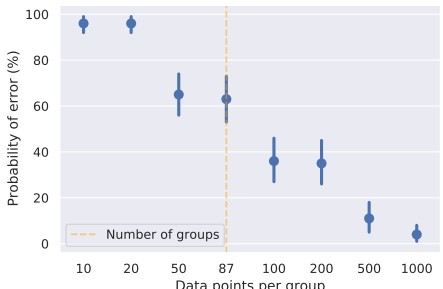 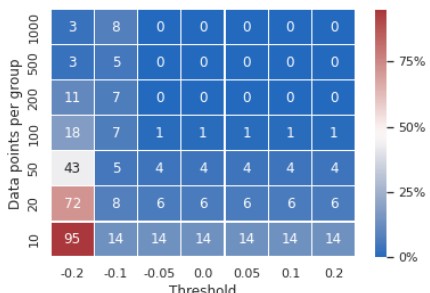

**Figure 2:** Plots generated using the semi-synthetic data generated using **HSLS** dataset Ingels et al. (2011). For each value $m$ of data points per group in the x-axis, we create a semi-synthetic dataset with $m$ data points per group. Then, we run 100 Monte Carlo simulations to estimate the BoP using the empirical estimator and then run the threshold test. The 95% confidence intervals were calculated via bootstrap using Seaborn v0.11.2 Waskom (2021) and the empirical probabilities of error. The group attributes are given in Table 2. **Left:** Error probability, with 95% confidence interval bars, for a hypothesis test using the threshold test with $\epsilon = -0.15$. **Right:** Heat map of the probability of error in hypothesis test using the threshold test with $\epsilon$ given on the x-axis and with the number of data points per group given on the y-axis.

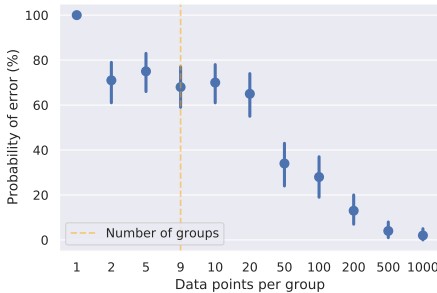 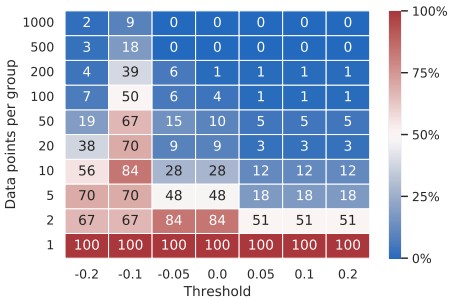

**Figure 3:** Plots generated using the semi-synthetic data generated using **COMPAS** dataset Angwin et al. (2016). The plots were generated using the same method described in Figure 2. The group attributes are given in Table 2. **Left:** Error probability, with 95% confidence interval bars, for a hypothesis test using the threshold test with $\epsilon = -0.15$. **Right:** Heat map of the probability of error in hypothesis test using the threshold test with $\epsilon$ given on the x-axis and with the number of data points per group given on the y-axis.

# E    Author Contributions

**L. Monteiro Paes.**    Under the supervision and collaboration of F.P. Calmon and B. Ustun, L. Monteiro Paes proved all the theoretical results in this paper and wrote the codes to generate the experiments featured in the images of the paper. In addition, in collaboration with all the authors, L. Monteiro Paes contributed to mathematical formulations and paper writing.

**C.X. Long**    contributed to mathematical formulations, proofs, paper writing, and numerical experiments.

**F.P. Calmon and B. Ustun**    designed the research and problem formulation and contributed to mathematical formulations, proofs, paper writing, and numerical experiment design.

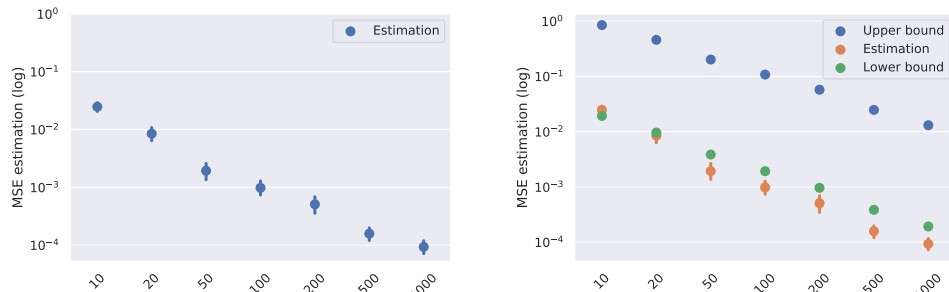

**Figure 4:** Plots generated using the semi-synthetic data generated using **HSLS** dataset. For each value $m$ of data points per group in the x-axis, we create a semi-synthetic dataset with $m$ data points per group. Then, we run 100 Monte Carlo simulations to estimate the BoP using the empirical estimator and then calculate the MSE. The 95% confidence intervals were calculated via bootstrap using Seaborn v0.11.2 and the empirical MSE. The group attributes are given in Table 2. **Left:** MSE of the BoP estimation for different values of samples per group in log scale. **Right:** Also in log scale, MSE of the BoP estimation, but additionally, it shows the lower and upper bound in Theorem 2

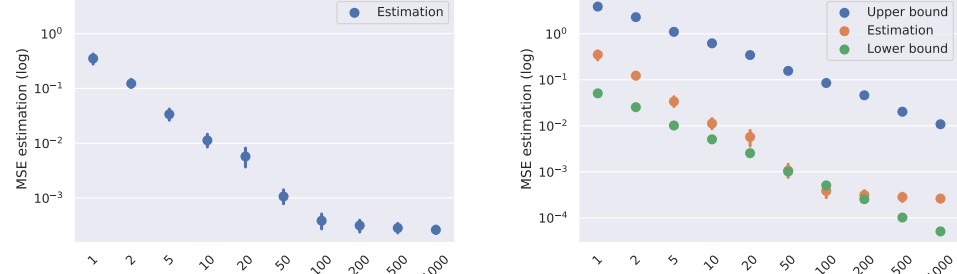

**Figure 5:** Plots were generated using the same semi-synthetic data generated using **COMPAS** dataset and the same technique as in Figure 4. **Left:** MSE of the BoP estimation for different values of samples per group in log scale. **Right:** Also in log scale, MSE of the BoP estimation, but additionally, it shows the lower and upper bound in Theorem 2