# OpenReview forum: "On the Epistemic Limits of Personalized Prediction"
_NeurIPS.cc/2022/Conference — NeurIPS 2022 Accept_

### Official Review · Reviewer_h2Fj · 2022-07-10

**Rating:** 4
**Confidence:** 4
**Ethics Flag:** Yes
**Soundness:** 3 good
**Presentation:** 3 good
**Contribution:** 3 good

**Summary:**

The authors propose a new metric, benefit of personalization (BoP), to quantify the minimum gain in accuracy by using group attributes across all subgroups for a prediction model. They also develop a hypothesis testing framework to test fair-use of group attributes. They develop an information-theoretic limit on the error probability for any hypothesis test which is auditing a model for fair-use. They also characterize the statistical limit on the MSE for any estimate of BoP. Finally they conduct experiments to demonstrate the behavior of the probability of error in testing for fair-use

**Questions:**

The major comment is on the definition of the benefit of personalization metric. I agree with Definition 1 (Fair Use) where we would expect for each $j$ the accuracy of the personalized model should be *equal* and/or greater than that of the group blind model. However, when defining BoP, taking the min over j, a lot of the information is actually lost.

Especially in the corner case where we benefit every group but one where we get equality, the gamma turns out to be zero since we are taking the minimum. Hence from the recommendation point of view, forcing the test on BoP > epsilon would not let us deploy any of these models where we are benefitting all group lest one (or a few).

In such situations, it is much more preferable to deploy a new personalized model to the other groups while keeping the base model for that group. Being able to quantify such a metric where are not actively harming a group, but making it better for the majority of others should be a good forward step.

Can the authors throw some light on this nuance and actively discuss how to change the metric towards this direction? Also curious how the Information-Theoretic results will change with such a new metric.



**Ethics Review Area:**

["Discrimination / Bias / Fairness Concerns"]

**Limitations:**

The major limitation is on the recommendation based on the metric defined in this paper. The details are highlighted in the question above. But following the recommendation would make us miss a big corner case where in practice it is not possible to improve accuracy for each  group but possible for a wide range and keeping it same for others.


**Strengths And Weaknesses:**

The paper is well written and is easy to follow. The associated theoretical results look good.

---

> ### Author Response · Authors · 2022-08-02
> **Response to Reviewer h2Fj [Part 1]**
>
> Thank you very much for the review and questions!
> ### Definition of BoP – Are we losing too much information by taking the min?
>
> Taking the min is necessary because it ensures fair use of users’ personal information prior to data collection and sharing. Collecting and revealing personal data incurs a cost, particularly in healthcare applications (consider invasive medical tests, privacy concerns on revealing HIV-status). Before users collect and reveal their data to a model, they should be informed of the BoP. When the BoP is negative or small, the user can use this information to weigh the costs of collecting/sharing data with the model holder. Being able to estimate the BoP across all groups (i.e., taking the $\min$ in our formulation) ensures that the potential accuracy gain (or lack theoreof) can be communicated to all model users, regardless of their group affiliation.
>
> This paper answers when the reliable estimation of the $\min$ is (im)possible. If the number of samples allows the BoP to be reliably estimated across all groups, then the next step is to decide which groups should use a personalized model or not. Note that the BoP need not be positive, but it must be estimated accurately. However, if the BoP cannot be precisely estimated, then the decision of deploying a personalized model becomes challenging: even though we may estimate a gain in personalization for some groups, it can be (information-theoretically) infeasible to tell if that gain is statistically significant, or simply a fluke due to lack of samples or too many groups. Our bounds precisely capture the trade-off between number of groups, sample size, and gain in personalization.
>
> In fact, we encourage ML practitioners to report per-group benefit for all groups, but only after using our bounds to check that BoP (minimum per-group benefit) can be reliably estimated. We have added this discussion in lines 353-360 in the revision for clarification.
>
> ### Is the paper suggesting one should not deploy models that benefit all but one group?
>
> No, this is not what we are suggesting, and we thank the reviewer for raising this point of confusion. Our goal is to understand under which conditions (i.e., sample size, number of groups) fair use can be verified in a statistically significant manner. Regardless if the BoP is positive (all groups benefit from personalization) or negative (at least one group is harmed), it must first be estimated accurately, followed by a decision on which groups should use the personalized model.
>
> If sample and group size allow accurate estimation of the BoP, then the next step is to decide how personalized models should be deployed and to report the benefit per group. For example, if BoP is positive, then all groups benefit, and personalized models can be used without harm across groups. On the other hand, if BoP is negative, then certain groups must revert back to the non-personalized model. Deciding which groups should do so requires first estimating the BoP with high confidence. Otherwise, as our results show, a perceived accuracy gain (or loss) from personalization may be the result of statistical fluctuations that depend on dataset size and number of groups. In summary, in order to decide which groups should use a personalized model, one must first estimate the BoP.
>
> The revised paper now includes mention of per-group benefit and elaborates more on model deployment and feature selection in the final section. In short, whether the model benefits a group, as measured by per group benefit, needs to be estimated from data, and our bounds prove the threshold for which the estimation of the $\min$ benefit (BoP) becomes unreliable in terms of the number of groups or sample size. Hence, by considering the reliable estimation of the minimum, our bounds are simultaneously answering the question of whether the benefit of personalization for any group can be reliably determined.
>
> Continued...

---

> > ### Author Response · Authors · 2022-08-02
> > **Response to Reviewer h2Fj [Part 2]**
> >
> > ### Corner case – benefit all groups but one for which there’s equality
> > This specific corner case is actually covered in our hypothesis test framework. In Section 3.1, a personalized model is adopted if BoP $\ge \epsilon$, where $\epsilon$ can be 0.
> >
> > A related scenario that may help elucidate this question is when the personalized model is harming at least one group – i.e. the empirical BoP is negative. Our theoretical framework and bounds are still directly applicable here! In this case, the estimated BoP can indeed help inform for which groups to deploy personalized models and for which to revert to the baseline model, but only after verifying the reliability of the estimated BoP using our bound on estimation (Theorem 2).
> >
> > This discussion is included in Section 5 lines 353-355 of the paper. For a similar discussion, please refer to our reply to Review 2nKW’s question “Can we apply error bounds to identify a smaller subset of group attributes that produce lower BoP?”
> >
> > ### Being able to quantify such a metric where we are not actively harming a group, but making it better for the majority of others should be a good forward step.
> > BoP does exactly that when reliable estimation is possible. We highlight that our goal is to understand when the BoP can be reliably estimated across groups. As mentioned above, estimating the BoP ensures that we can estimate the per-group accuracy gain, and thus identify which groups should revert to a non-personalized model. Please, refer to Section 5 lines 353-360 for a more detailed discussion.
> >
> > Hope this clarifies your questions! Please let us know if there are any more concerns, and we are happy to address them here and in the final version of the paper!

---

> > > ### Author Response · Authors · 2022-08-05
> > > **Thanks again for your review and time**
> > >
> > > Thanks again for your thoughtful review! We believe that we have addressed all of your concerns and questions in our response above. We would love to receive any additional feedback you may have. Do you have any follow-up questions? We are excited to engage in further discussions this week! Please let us know.
> > >
> > > Thank you very much, and we look forward to hearing from you.

---

> > > > ### Author Response · Authors · 2022-08-08
> > > > **Thank you again for your time and feedback**
> > > >
> > > > Thank you again for your time and feedback!
> > > >
> > > > We're writing since we are now nearing the end of the author-reviewer discussion period but have not heard from you yet.
> > > >
> > > > If you have other concerns, please let us know soon. If we have addressed your questions, we would appreciate it if you would kindly consider raising your score.

---

### Official Review · Reviewer_bqZa · 2022-07-13

**Rating:** 5
**Confidence:** 3
**Soundness:** 3 good
**Presentation:** 3 good
**Contribution:** 3 good

**Summary:**

The authors propose a new fairness notion called benefit or personalisation(BoP), in which a personalized classifer (i.e. a classifier that is specific to subgroups) is compared to a classifier that does not use group-membership for prediction. The authors propose a hypothesis testing framework and give a lower bound on the sample complexity for successful testing. The paper then shows tight upper and lower bounds on the MSE estimation of BoP.

**Questions:**

How do you pick the baseline model? How do you make sure that the baseline model is not already biased against certain subgroups?
Can we have personalization only for groups big enough?

**Limitations:**

I raised some concerns in the questions part. I think some warning about choosing the baseline model is required.

**Strengths And Weaknesses:**

strength:

I like the idea of benefit of personalisation. As a fairness consideration it makes sense and I have not seen it in the literature.
I agree with the opinion that this should be a consideration taken into account when deploying personalized predictions

I think it was a great idea to make sample sizes concrete and comparing them to the number of people currently alive. I think this made the point very effectively and might be a helpful method when analyzing sample complexities in the context of fairness or human-centric ML in general.
The emphasis on giving a clear message to practitioners that derives from these theorems was appreciated.

The bounds of theorem 2 are tight, making them quite informative.

several limitations of the paper are addressed

the paper is overall well written. The related work section is sufficiently detailed.

weaknesses:

It is unclear to me what h_0 is. I think this is a crucial thing to discuss. A practitioner might devise a model that (while not explicitly using group information) might be bad on certain subgroups as a comparison. Using such a model as a bad-faith comparison is currently not discouraged or addressed in the paper. I think in practice it might also be unclear how to choose the baseline model.

One might come up with a model that uses the baseline model for certain groups in which there aren't enough samples. If the models are the same for those groups this might be checked deterministically, not needing any samples for this group. Thus testing could be more efficient in such cases. Groups might also be found via hierarchal clustering to account for group-size. This might solve the issue of having to test a number of groups which is exponential in the number of features. I think these alternative views should be highlighted in the paper. Assuming that each subgroup needs to be viewed independently and they cannot be grouped together might be misleading.

guarantees are worst-case in the distribution; distributions in the real world might be nicer (this concern was raised by the authors themselves)
----

I think this line of work has a lot of potential and there are many parts I like about this paper. Ultimately I think there are still some unanswered questions at this point that I would wish the paper to address in its final version.

---

> ### Author Response · Authors · 2022-08-02
> **Response to Reviewer bqZa**
>
> Thank you very much for the thoughtful review and insightful comments! We also appreciate the kind feedback regarding presentation, technical soundness, and novelty.
>
> ### What is $h_0$ and how to pick the base model?
> $h_0$ is a model trained without using group attributes. For example, in medical applications, the baseline model is usually an existing diagnostic tool that uses routinely available medical information, rather than one that uses additional group-denoting features (defined in lines 19-22). More generally, we assume $h_0$ is a baseline model trained using empirical risk minimization on a dataset without group attributes. We clarify this point in lines 144-148. We further remark on the dangers of bias in the base model below.
>
> ### Can we have personalization only for groups big enough?
> Yes, “fair personalization” (i.e. personalization that satisfies fair use for all groups) cannot be ensured when there are too few samples or too many groups.  To ensure fair use of personal information for all groups (Definition 1), we need to be able to estimate BoP reliably. Our bounds prove that, when we have too few samples (or too many groups), the estimates of the BoP and per group gain in accuracy become unreliable: by Theorem 2, the variance of the estimate becomes large. Note that this is a phenomena that occurs across groups: even if the number of samples per group is maintained, as the number of groups increases the likelihood that the gain in personalization is incorrectly estimated for at least one group increases.
>
> If the BoP cannot be precisely estimated, then the decision of deploying a personalized model becomes challenging: even though we may estimate a gain in personalization for some groups, it can be (information-theoretically) infeasible to tell if that gain is statistically significant, or simply a fluke due to lack of samples or too many groups. In such situation, one should either reduce the number of groups or use the baseline model for some groups. We have included this discussion in lines 353-360.
>
> ### How do you make sure the baseline model is not biased against certain subgroups?
> Great point! BoP is not a replacement for group-fairness metrics such as Equalized Odd and Equal Opportunity. Indeed, group fairness metrics should still be estimated when applicable. BoP is a complementary fairness metric that takes on the data holder/stakeholder’s perspective to ensure fair use of users’ personal information. Simply measuring the BoP may mask imbalanced performance across groups. Conversely, group fairness does not capture the gain (or loss) in using group attributes. We have added this warning to the paper in Section 1.1 (lines 89-91) and Section 5 Limitations (lines 345-348).
>
> ### Warnings about choosing the baseline model.
> Thanks for the suggestion! We have added the guidelines on the choice of the baseline model, including warnings against bad-faith baselines and checking for other relevant group fairness metrics, to both Section 2 (lines 147-148) and Section 5 Recommendations of the main paper (lines 353-360).
>
> ### Alternative views on clustering groups for efficient testing.
> Thanks for the comments on how to improve the efficiency of testing. We incorporated some ideas into recommendations in Section 5 lines 353-360 when one group has substantially fewer data points than others.
>
> We would like to note that though the approaches differ in efficiency, our results on the reliability of the tests still hold. Our bound (see Theorem 1 and Corollary 1) characterizes when testing becomes unreliable due to the probability of error growing too large in terms of the number of attributes and sample size. This is an information-theoretic bound that holds regardless of the testing algorithm.
>
> Hope this clarifies your main questions – thanks again. Please let us know if there are any more concerns, and we are happy to address them here and in the final version of the paper!

---

> > ### Author Response · Authors · 2022-08-05
> > **Thanks again for your review and time**
> >
> > Thanks again for your thoughtful review! We believe that we have addressed all of your concerns and questions in our response above. We would love to receive any additional feedback you may have. Do you have any follow-up questions? We are excited to engage in further discussions this week! Please let us know.
> >
> > Thank you very much, and we look forward to hearing from you.

---

> > > ### Author Response · Authors · 2022-08-08
> > > **Thank you again for your time and feedback**
> > >
> > > Thank you again for your time and feedback!
> > >
> > > We're writing since we are now nearing the end of the author-reviewer discussion period but have not heard from you yet.
> > >
> > > If you have other concerns, please let us know soon. If we have addressed your questions, we would appreciate it if you would kindly consider raising your score.

---

### Official Review · Reviewer_2nKW · 2022-07-17

**Rating:** 6
**Confidence:** 4
**Soundness:** 4 excellent
**Presentation:** 3 good
**Contribution:** 3 good

**Summary:**

The paper contributes to fairness literature by proposing a new metric, the benefit of personalization (BoP), to measure the benefit of including group attributes in a binary prediction model. The metric measures the worst-case improvement across subgroups, which is attractive in settings where parity is not a reasonable metric, but "do no harm" is an appropriate metric, such as healthcare. The paper considers two settings in which the metric may be used, hypothesis testing and estimation, and provides general characterizations on the reliability of the metric. Specifically, the paper bounds the worst-case error in hypothesis testing and the MSE of the BoP estimator relative to the number of sub-groups and the number of samples per subgroup. Intuitively, they show the BoP metric is more reliable as the number of samples per subgroup increases. These results are then used to characterize general rules of thumb on when BoP estimator is reliable for estimation or testing. Numerical experiments are provided throughout to highlight how well the BoP metric performs empirically in both testing and estimation settings.

**Questions:**

1. Is it possible that a smaller subset of group attributes produces positive BoP and all group attributes produces a negative BoP? For example, I can imagine a scenario that for 10 group attributes, there may exist a subset of the group attributes that improve classification accuracy of the machine learning model but including all 10 may decrease the classification accuracy of a subgroup.

2. As a follow up to 1., is it possible to comment on the reliability of BoP in the setting that you perform the hypothesis test on different subsets of the group attributes? Can you just apply existing results in multiple hypothesis testing to do so? This would potentially allow you to identify the smaller subset of group attributes in question 1.

3. Similar to 2., does the MSE result (Theorem 2) hold uniformly over different subsets of group attributes?

4. Does focusing on accuracy for BoP guarantee improvements in the false negative and false positive rate? If not, can the BoP metric be modified to measure only false negatives or only false positives rather than accuracy?

5. For Figure 2 Right, do you mean higher values of $\| \epsilon \|$ have higher tolerance for mistakes in the test?

6. In Corollary 2, is there a reason that the 5% threshold was chosen as the default benchmark? It may be important to clarify the choice as it directly affects the maximum number of binary group attributes that still guarantee reliable estimation of BoP.




**Limitations:**

The authors mainly address limitations related to the drawbacks of their worst-case results in settings where stronger assumptions may be made on the underlying data distribution. There isn't a discussion of the negative societal impact of their work, but it is unclear if there is a specific society impact they can discuss. Perhaps further discussion can be made on why accuracy was chosen as the benchmark metric rather than something similar such as the false positive rate or false negative rate.

**Strengths And Weaknesses:**

This is a well-written paper that provides an interesting metric, BoP, to address a rapid expansion of personalization in machine learning. The paper does a good job of overviewing the setting and the motivation for developing the metric and provides useful characterizations of the metric. By considering worst-case bounds, the paper can construct conservative rules outlining when the metric can be used to make the critical decision of whether or not to introduce group attributes into personalization machine learning models. The paper seems technically sound, and the results are well presented.

The main drawback of the paper may lie in the significance of the BoP itself, as it only considers the worst-case improvement of the machine learning models, which may be overly conservative. The BoP metric thus can only make the simple and coarse decision of introducing group attributes or not introducing group attributes to a model. In contrast, a potentially more exciting problem could be deciding which subgroups utilize and which subgroups do not utilize the machine learning model with group attributes. This example reflects a more granular version of the problem BoP tries to solve and captures the fair use problem similarly.

---

> ### Author Response · Authors · 2022-08-02
> **Response to Reviewer 2nKW [Part 1]**
>
> Thank you very much for the thoughtful review and questions!
>
> ### "The main drawback (...) in the significance of the BoP itself, as it only considers the worst-case improvement of the machine learning models, which may be overly conservative."
>
> We thank the reviewer for raising this important point. We would like to highlight two aspects of why we chose the $\min$ in our definition of BoP. We have added this discussion to Section C.1 of the Supplementary Material.
>
> First, please note that we are considering the problem of reliably estimating the BoP, regardless if it is positive (all groups benefit) or negative (at least one group is harmed). We are not advocating for deploying personalized ML models only if every single group receives a gain in accuracy.
>
> If the number of samples allows the BoP to be reliably estimated across all groups, then the next step is to decide which groups should use a personalized model or not. However, if the BoP cannot be precisely estimated, then the decision of deploying a personalized model becomes challenging: even though we may estimate a gain in personalization for some groups, it can be (information-theoretically) infeasible to tell if that gain is statistically significant, or simply a fluke due to lack of samples or too many groups. Our bounds precisely capture the trade-off between number of groups, sample size, and gain in personalization.
>
> Second, the BoP is designed to empower data holders, i.e. users of the ML model. A user's group attribute may not be known a priori and may incur a collection cost – particularly in healthcare applications. For example, group attribute collection can require invasive procedures (e.g., blood draw to determine HIV status) or require that the user reveal private information to the model holder (e.g., substance use). In such cases, it is essential that the collection and disclosure of group attributes ensure a gain in accuracy, regardless of what the value of the attribute is. Alternatively, the user should be informed if there is a chance that they may not receive a prediction gain with the disclosure of a group attribute (i.e., non-positive BoP). This information can enable the user to make an informed decision to collect and disclose personal data (e.g., if a blood draw merits the gain in diagnostics accruacy by a ML model). Either way, if the value of a group attribute is not known a priori, we must reliably estimate the BoP for all groups – this is exactly what our bounds capture.
>
> ### Can we apply error bounds to identify a smaller subset of group attributes that produce lower BoP?
>
> Yes, error bounds indeed help in identifying a smaller subset of group attributes with either higher or lower BoP – we provided examples of identifying the group that is most harmed in the SM (see the table below, SM Section D.3, and corresponding code). Our bounds characterize the maximum number of attributes for which we can accurately estimate the BoP.  When the number of attributes is less than that, one can safely calculate the empirical BoP, identify the groups that are harmed by personalization the most, and proceed from there. On the other hand, if one has too few samples for certain groups, or equivalently, too many groups, the empirical estimation of the benefit becomes unreliable: by Theorem 2, the variance of the estimate becomes too high. In this case, our bounds can serve as a guide on how the number of groups can be reduced in order to enable reliable estimation of the personalization gain. We have added this discussion to Section 5 line 353-355.
>
> In the corresponding codes, we have provided a tool and examples on how to calculate the empirical BoP and identify the group producing the worst benefit. Table 3 in the supplementary material shows the result (see codes for exact implementation). For example, in our experiment on COMPAS, (Asian, Male) has the minimum per-group benefit of -9%. After using the error bounds to confirm such estimation is statistically reliable, the tool can be used to inform feature selection and whether the personalized model should be applied to this group.
>
> ### Reliability of hypothesis test and MSE results on different subsets of the group
>
> Yes, both bounds hold when different subsets of the group attributes are tested. For a smaller subset, we get a less personalized model and higher confidence in auditing, as reflected in favorable bounds for both testing error probability (see Theorem 1) and MSE of the estimated BoP (see Theorem 2).
>
> When we require multiple hypothesis tests to hold simultaneously, testing separately would have the same effect as taking the union of the attributes and doing one test, because our bounds do not differentiate on the specific attribute being tested.
>
> Continued...

---

> > ### Author Response · Authors · 2022-08-02
> > **Response to Reviewer 2nKW [Part 2]**
> >
> > ### Does focusing on accuracy for BoP guarantee improvements in false negative/false positive rate? Can BoP be modified to measure only FPR/FNR rather than accuracy?
> >
> > Yes to both questions! We appreciate this comment and have included the discussion in Section 2.2 the (lines 167-169), Section 5 of the main paper, as well as SM Section B lines 40-48.
> >
> > First, when we improve the accuracy we get a better FPR or FNR (not exclusive or). This is a consequence of the fact that FPR, FNR, and accuracy are linearly related as:
> >
> > $\mathrm{accuracy}= \Pr (Y = h)$
> >
> > $ = 1 - \Pr(Y \ne h)$
> >
> > $= 1 - ( \Pr(h = 1 | Y= 0)\Pr(Y = 0) + \Pr(h = 0 | Y= 1)\Pr(Y = 1)  )$
> >
> > $= 1 - (\text{FPR} \cdot \Pr(Y = 0) +\text{FPR} \cdot \Pr(Y = 1)  )$
> >
> > Note that $\Pr(Y = 0)$ and $\Pr(Y = 1)$ are fixed.
> >
> > More importantly, our theorems still hold if we replace accuracy with the false positive rate or false negative rate. In particular, our proofs rely on the observation that accuracy (of a model on a data point) can be written as a Bernoulli random variable where 1 denotes an accurate prediction and 0 otherwise, and characterize the two hypotheses in terms of the means of these distributions (see SM lines 15-19). Analogously, FP and FN of a model can also be represented by a Bernoulli random variable where 1 denotes an FP/FN and 0 denotes a TP/TN.The proof then carries out directly. We will add this observation to the SM Section B lines 40-48.
> >
> > ### Why accuracy rather than false positive rate/false negative rate?
> >
> > Great question! We initially focused on "error rate" since it is so widely used. As mentioned above, the results generalize directly to class-specific error metrics like TPR/FPR, and we have added this important observation of our results to Section 2.2 the (lines 167-169), Section 5 of the main paper, as well as SM Section B lines 40-48.
> >
> > ### For Figure 2 Right, do you mean higher values or have higher tolerance for mistakes in the test?
> >
> > In this figure, we aim to show that for $\epsilon$ that is closer to the true value of the BoP the confidence of the hypothesis test decreases, as we expect. For Figure 2 we have a classifier with $BoP = -10$% hence the confidence of the hypothesis test increases with the distance from $-10$%. We observe the same phenomenon in other classifiers and different values of $\epsilon$ (see Supplementary Material Figures 2 and 3 Right).
> >
> > ### In Corollary 2, is there a reason that the 5% threshold was chosen as the default benchmark?
> > We provide a general characterization in the first part of Corollary 2. We then choose $5$% in the second part as an example. The motivation is that 5% decrease in accuracy is already significant in many applications. Our result is robust to changes in the threshold ($\delta$). For instance, if we choose $\delta = 20$%, the limit would be 28 attributes, an increase of only 3 attributes. A practitioner’s choice of delta depends on the application and context, and Eq. 9 provides the limit for your choice of delta.
> >
> > We hope this clarifies your questions. Please let us know if there are any more concerns, and we are happy to address them here and in the final version of the paper!

---

> > > ### Author Response · Authors · 2022-08-05
> > > **Thanks again for your review and time**
> > >
> > > Thanks again for your thoughtful review! We believe that we have addressed all of your concerns and questions in our response above. We would love to receive any additional feedback you may have. Do you have any follow-up questions? We are excited to engage in further discussions this week! Please let us know.
> > >
> > > Thank you very much, and we look forward to hearing from you.

---

> > > > ### Author Response · Authors · 2022-08-08
> > > > **Thank you again for your time and feedback**
> > > >
> > > > Thank you again for your time and feedback!
> > > >
> > > > We're writing since we are now nearing the end of the author-reviewer discussion period but have not heard from you yet.
> > > >
> > > > If you have other concerns, please let us know soon. If we have addressed your questions, we would appreciate it if you would kindly consider raising your score.

---

> > > > > ### Comment · Reviewer_2nKW · 2022-08-09
> > > > > **Thank you for the response**
> > > > >
> > > > > I appreciate the response and feel that they adequately addressed the questions I had about the paper. The additions highlight that the BoP can be potentially constructed for different metrics highlighting its potential generality and further clarifies its focus on "considering the problem of reliably estimating the BoP".  The main insights of BoP seem to be that it should used to make decisions in selecting which models to explore, "our bounds can serve as a guide on how the number of groups can be reduced in order to enable reliable estimation of the personalization gain", but not for model selection as they state, "We are not advocating for deploying personalized ML models only if every single group receives a gain in accuracy." The authors seem to suggest that reliable BoP implies that users can make model selection decisions based on accuracy gain/loss, but without additional theoretical results it seems unclear to me if that is true. As a result, I will keep my current score.

---

> > > > > > ### Author Response · Authors · 2022-08-09
> > > > > > **Major clarification**
> > > > > >
> > > > > > We need to clarify something while we still can: the primary purpose of BoP is **not** to “select which models to explore.” It's true that BoP could be used in this way, though we agree with you that this would require further work.
> > > > > >
> > > > > > **However, this is not the primary purpose, nor the “main insight”, of this work.** The primary purpose of BoP is to flag a model where personalization would lead to unnecessarily inaccurate predictions for at least one group among all groups who provide personal data. By defining BoP in this specific way, we can characterize when personalization works for all groups who provide personal data. Stepping back, this is the primary purpose of the metric and the main insight of our paper -- i.e., a definitive result on when we can/cannot tell whether personalization benefits each group who provides personal data.
> > > > > >
> > > > > > To be clear, our results are connected to model selection in a very narrow sense. In particular, the “model selection” that we were referring is between two models: one personalized (by using group attributes), and one not personalized. Ideally, each group should know the benefit (in accuracy) of using a personalized model, and then check if this merits reporting personal data. Thus, one must accurately estimate the benefit/gain in accuracy for each group first. The statistical limits of this estimation task are precisely what our results characterize. This is different from traditional model selection -- e.g., when choosing between model classes or tuning hyperparameters. Again, we make no claims regarding the latter in our paper.

---

### Review · Ethics_Reviewer_FcY3 · 2022-08-03

**Recommendation:**

I would like the authors to include a more elaborate discussion about the two points mentioned above.

**Ethical Issues:**

Yes

**Ethics Review:**

The paper presents a new metric in the context of algorithmic fairness, which considers improvement in prediction accuracy across all sub-groups (according to sensitive attributes) as a measure of fair use of these attributes by the prediction algorithm.

One main concern, also raised by some of the technical reviews, is regarding the selection of the baseline. As the criterion the authors suggest relies heavily on this selection. Poor selection of this baseline could result in very low performance on specific groups to begin with, allowing the improvement criterion to be met on these groups relatively easily, where the resulting eventual level of accuracy on these groups are still significantly lower than on other groups. A more elaborate discussion around this by the authors is desirable.

Another point is regarding the usage of the group with the smallest benefit as the measure of fair usage, and the preference of solutions which raise this minimum, but not necessarily the overall improvement in the population. This is especially relevant, as in practice, it could be easier in some cases to improve accuracy on a part of the groups, while not causing harm in the others. In the context of medical uses, any such improvement could be crucial (it is possible, for example, that some new treatment is proven to be highly successful for some groups, but not for others, despite genuine efforts to provide a treatment which is successful for all of the groups). On the other hand, the definition suggested by the authors has the desirable property that it incentivizes improving accuracy for all of the groups in the population, which would not allow focusing deliberately only on some groups, leaving others behind. I would appreciate a discussion by the authors on these alternatives in defining the measure of fair usage, and the contexts in which they believe they should be preferred.

---

### Review · Ethics_Reviewer_j83v · 2022-08-05

**Recommendation:**

Given the author response and updated pdf, I do not see a reason to reject this paper on ethics grounds alone.

However, after reading the paper and skimming some earlier related works by Ustun and coauthors, I did find myself wondering about whether subgroup accuracy (or even other prediction statistics such as FNR/FPR) is a suitable stand-in for subgroup preference and/or harm. Is it really the case that increasing accuracy (according to the target label Y) leads to a preferred model for the subgroup in question? This brings to mind troublesome applications of risk assessment trained on proxy targets (e.g. re-arrest rate for risk of recitivate, or health care costs for risk of adverse health outcome), where increasing accuracy may not be a desirable model outcome for certain groups or individuals. For example if there is a sampling bias or group-specific noise profile on the targets, there could be settings where having access to the group label affords greater “accuracy” in the prediction task by simply predicting risk/no risk based on group membership, which could be very harmful.

Personally, I think the paper would be improved by a discussion of this subtlety, which I couldn't find in the cited papers (correct me if I missed something). On the other hand, I wasn’t assigned as a reviewer to this paper so the AC may reasonably choose not to consider this point when making an accept/reject recommendation.


**Ethical Issues:**

Yes

**Ethics Review:**

Reviewer h2Fj brought up a concern around how the proposed benefit of personalization (BoP) would influence group-aware models being deployed (or not deployed) in practice. As an example, could there exist a scenario where a group-specific model was not deployed because of the BoP estimate, namely if it was determined to benefit all subgroups except one?

---

### Author Response · Authors · 2022-08-02
**Thank you for your constructive comments!**

We thank all reviewers for their time and feedback! We are glad all reviewers recognized the novelty and positive aspects of our work, describing our paper as “technically sound, and the results are well presented” (Reviewer 2nKW), “(a)s a fairness consideration it makes sense and I have not seen it in the literature”, “very effectively (...) when analyzing sample complexities in the context of fairness or human-centric ML” (Reviewer bqZa) and “well written”, “associated theoretical results look good” (Reviewer h2Fj). In particular, we are glad that all reviewers assigned a 4 or 3 to soundness, presentation, and contribution!

We appreciate the reviewers’ thoughtful input! We believe we have addressed all critical points in the replies below. We have also incorporated many suggestions in the revision of the paper and SM (marked in blue). Please follow up with us, and we welcome any feedback that can further strengthen our paper!

---

### Meta-Review · Area_Chair_uS8W · 2022-08-24

**Recommendation:** Accept
**Confidence:** Certain

**Metareview:**

Thank you for submitting your paper to NeurIPS! This paper proposes a new metric, the benefit of personalization (BoP), to verify fair use of predictive models that have varied subgroup-level performance. Reviewers raised a variety of concerns about how the approach should be used in real-world contexts. I believe that these concerns were satisfactorily addressed by the authors and therefore I recommend acceptance (despite the fact that the most negative reviewer failed to engage in the discussion and update their score). There were additional concerns raised by the ethics reviewers that authors did not have a chance to address, but that I fully expect the authors to incorporate in the body of the paper (in addition to the concerns flagged by the reviewers).

Let me summarize the concerns raised in the ethics reviews. First, the choice of the baseline is critical -- poor selection of the baseline could result in very low performance on specific groups to begin with, allowing the improvement criterion to be met on these groups relatively easily, despite potentially large disparities in performance across groups. Second, fairness can be context-dependent, e.g. in clinical trials, improvements in any subgroup is desirable even if some subgroups do not benefit; on the other hand, the proposed definition aligns incentives to improve performance for all subgroups. These trade-offs are crucial and merit discussion. Finally, there was concern regarding whether subgroup accuracy is a suitable stand-in for subgroup preference and/or harm, i.e. can increasing accuracy (according to the target label Y) lead to worse outcomes for a subgroup? See, for example, work by Ustun et al. on risk assessment trained on proxy targets (re-arrest rate for risk of recidivism, or health care costs for risk of adverse health outcomes), where increasing accuracy may not be desirable (e.g. due to sampling bias). I strongly urge the authors to discuss these issues (as they have partly done already) and highlight them in prominent portions of the paper.

**Award:**

No

---

### Decision · Program_Chairs · 2022-09-14

Accept